# A₂ₐR eGFP reporter mouse enables elucidation of A₂ₐR expression dynamics during anti-tumor immune responses

Kirsten L. Todd [1,2,7] ✉, Junyun Lai [1,2,7], Kevin Sek [1,2], Yu-Kuan Huang [1,2], Dane M. Newman[2,3], Emily B. Derrick [1,2], Hui-Fern Koay [4,5], Dat Nguyen[1,2], Thang X. Hoang[1,2], Emma V. Petley[1,2], Cheok Weng Chan[1,2], Isabelle Munoz [1,2], Imran G. House[1,2], Joel N. Lee[1,2], Joelle S. Kim[1,2], Jasmine Li[1,2], Junming Tong[1,2], Maria N. de Menezes[1,2], Christina M. Scheffler [1,2], Kah Min Yap[1,2], Amanda X. Y. Chen[1,2], Phoebe A. Dunbar[1,2], Brandon Haugen[1,2], Ian A. Parish [1,2], Ricky W. Johnstone [2,3], Phillip K. Darcy [1,2,6] & Paul A. Beavis [1,2] ✉

There is significant clinical interest in targeting adenosine-mediated immunosuppression, with several small molecule inhibitors having been developed for targeting the A₂ₐR receptor. Understanding of the mechanism by which A₂ₐR is regulated has been hindered by difficulty in identifying the cell types that express A₂ₐR due to a lack of robust antibodies for these receptors. To overcome this limitation, here an A₂ₐR eGFP reporter mouse is developed, enabling the expression of A₂ₐR during ongoing anti-tumor immune responses to be assessed. This reveals that A₂ₐR is highly expressed on all tumor-infiltrating lymphocyte subsets including Natural Killer (NK) cells, NKT cells, γδ T cells, conventional CD4⁺ and CD8⁺ T lymphocytes and on a MHCII^hi CD86^hi subset of type 2 conventional dendritic cells. In response to PD-L1 blockade, the emergence of PD-1⁺A₂ₐR⁻ cells correlates with successful therapeutic responses, whilst IL-18 is identified as a cytokine that potently upregulates A₂ₐR and synergizes with A₂ₐR deficiency to improve anti-tumor immunity. These studies provide insight into the biology of A₂ₐR in the context of anti-tumor immunity and reveals potential combination immunotherapy approaches.

Adenosine is an immunosuppressive metabolite that modulates anti-tumor immunity and is produced from the degradation of adenine nucleotides by the ecto-enzymes CD73 and CD39 on tumor cells, stroma and fibroblasts or through direct export from the intracellular compartment of cells undergoing hypoxia and/or stress[1–7]. Adenosine binds to four known G-coupled receptors, A₁R, A₂ₐR, A₂ᵦR, and A₃R on immune cells with the majority of immunosuppressive effects thought to be mediated through A₂ₐR and A₂ᵦR. A₂ₐR and A₂ᵦR elicit immunosuppressive effects on tumor-infiltrating immune cells due to activation of adenylate cyclase, which results in increased intracellular

[1]Cancer Immunology Program, Peter MacCallum Cancer Centre, Melbourne 3000 VIC, Australia. [2]Sir Peter MacCallum Department of Oncology, The University of Melbourne, Parkville 3010, Australia. [3]Translational Hematology Program, Peter MacCallum Cancer Centre, Melbourne, Australia. [4]Department of Microbiology & Immunology, Peter Doherty Institute for Infection and Immunity, University of Melbourne, Melbourne, VIC, Australia. [5]Australian Research Council Centre of Excellence in Advanced Molecular Imaging, University of Melbourne, Melbourne, VIC, Australia. [6]Department of Immunology, Monash University, Clayton, Australia. [7]These authors contributed equally: Kirsten L. Todd, Junyun Lai. ✉e-mail: Kirsten.Todd@petermac.org; paul.beavis@petermac.org

cAMP. Pioneering studies by Sitkovsky and colleagues have demonstrated that extracellular adenosine potently modulates immune responses and antitumor immunity[2,3].

Accordingly, enhanced anti-tumor responses are observed in A$_{2A}$R deficient mice[3,5,8,9], an observation that led to the development of small molecule antagonists, which elicit anti-tumor immunity against solid tumors in mice[3,5,8,10–12]. A$_{2A}$R antagonists have been evaluated in clinical trials against a range of solid tumors including renal cell carcinoma, non-small cell lung carcinoma, prostate, breast, and head and neck cancers[13]. Moreover, preclinical investigations have highlighted that targeting the A$_{2A}$R either genetically or pharmacologically enhances the efficacy of immune checkpoint blockade[5,8,14] or adoptive cellular therapy[3,12,15–21]. Despite the intense interest in targeting this pathway, the mechanism of action is not completely understood. While A$_{2A}$R is known to be expressed on a range of immune cell types including CD8$^+$ T cells, CD4$^+$ T cells[22], Tregs[23], NK cells[24,25], NKT cells[26], and myeloid cells[27], this has predominantly been determined from mRNA analyses due to a lack of an effective antibody recognizing this receptor that is suitable for flow cytometry. Therefore, a detailed analysis of the immune cell subpopulations expressing A$_{2A}$R have not been determined and, in addition, the factors driving A$_{2A}$R expression both within tumors at baseline and in response to therapy remain largely unknown. To overcome this, we describe a transgenic mouse model enabling the quantification of A$_{2A}$R through the expression of a GFP reporter gene. Using this model enables the expression of A$_{2A}$R on immune cells within solid tumors, draining lymph nodes (DLN) and the spleen to be assessed, allowing for analyses of A$_{2A}$R expression at baseline and in response to immunotherapies (anti-PD-L1, anti-CTLA-4 and IL-18) or chemotherapy (carboplatin).

This reveals that A$_{2A}$R is highly expressed in NK, CD8$^+$, CD4$^+$, NKT, and γδ T cells within tumors, with NK cells and conventional T cells constituting more than 50% of all A$_{2A}$R positive cells. Relatively limited A$_{2A}$R expression is observed on myeloid cells within the tumor, apart from cDC2s, and expression is absent from tumor-infiltrating B cells. When comparing the expression of A$_{2A}$R in the tumor and DLNs, T cells in the tumor uniquely express significantly more A$_{2A}$R than their counterparts in the DLN. We observe that whilst A$_{2A}$R expression is elevated in activated tumor antigen-specific CD8$^+$ T cells in the DLN, A$_{2A}$R expression is fairly ubiquitous amongst CD8$^+$ T cell subpopulations within the tumor, regardless of their antigen specificity or differentiation status. Treatment of tumor-bearing mice with anti-PD-L1 alone or combined anti-PD-L1 and anti-CTLA-4 leads to the emergence of a population of PD-1$^+$A$_{2A}$R$^-$ CD8$^+$ T cells that correlate with therapeutic outcome. Transcriptomic analysis of A$_{2A}$R$^+$ cells reveals that the most apparent difference between A$_{2A}$R$^+$ and A$_{2A}$R$^-$ CD8$^+$ T cells is reduced expression of STAT5 target genes in A$_{2A}$R$^+$ cells, a phenotype that is restored following anti-PD-L1 treatment. Lastly, these studies reveal that IL-18 is a potent inducer of A$_{2A}$R expression and that the therapeutic activity of IL-18 is accentuated in the context of A$_{2A}$R-deficient mice. In summary, we report the development of a mouse model that provides insight into the biology of A$_{2A}$R expression in the context of anti-tumor immunity and reveals potential mechanisms of action and combinatorial approaches for A$_{2A}$R blocking therapies.

## Results

### GFP expression accurately maps A$_{2A}$R expression in A$_{2A}$R eGFP reporter mice

To characterize the A$_{2A}$R eGFP reporter mice and ensure that GFP was a bona fide readout of A$_{2A}$R expression we first examined the expression of GFP within the spleens of mice. Immunofluorescent analysis indicated detectable expression of GFP in the spleens of A$_{2A}$R eGFP reporter mice, particularly in the T cell-rich region of the arterioles (Fig. 1A). Flow cytometry analyses of splenocytes revealed that GFP expression was highest in NK cells that are known to express high

levels of A$_{2A}$R (Fig. 1B)[24,28] and CD8$^+$CD44$^+$ T cells, which is also consistent with a previous report[29] (Fig. 1C). FACS sorting of splenocytes from A$_{2A}$R eGFP reporter mice indicated that GFP$^+$ splenocytes expressed significantly higher levels of $A_{2A}R$ mRNA (Fig. 1D), confirming that GFP expression correlated with increased expression of the A$_{2A}$R. Consistent with previous reports indicating that activation of T cells through the TCR led to increased $A_{2A}R$ mRNA expression[30], stimulation of splenocytes with anti-CD3/anti-CD28 led to a significant increase in GFP expression by both CD8$^+$ and CD4$^+$ T cells (Fig. 1E). Moreover, elevated GFP expression was associated with increased expression of PD-1 and CD44, highlighting that A$_{2A}$R expression was associated with an activated T cell phenotype in vitro (Fig. 1F). To confirm that A$_{2A}$R expression in reporter mice was functional we evaluated their response to adenosine receptor stimulation. Suppression of TNF by NECA, a pan adenosine receptor agonist, and reversal of this phenotype by SCH58261, an A$_{2A}$R antagonist, was observed following activation of splenocytes from A$_{2A}$R eGFP reporter mice or wild-type controls (Fig. 1G). To assess the impact of tumor antigen recognition on A$_{2A}$R expression, we generated anti-Her2 CAR T cells from A$_{2A}$R eGFP reporter splenocytes in line with our previous work using anti-Her2 CAR T cells[16]. Generation of CAR T cells led to increased expression of GFP relative to naïve T cells (Fig. 1H) and coculture of anti-Her2 CAR T cells with AT-3-, E0771-, or MC38-Her2 expressing tumor cells led to a further and significant induction of GFP in both CD8$^+$ and CD4$^+$ CAR T cells (Fig. 1I).

### A$_{2A}$R is expressed in a broad range of immune cells within the tumor microenvironment

A$_{2A}$R expression has been reported on a broad range of immune cells including conventional αβ T lymphocytes and NK cells but the expression of this receptor in the context of the tumor microenvironment has not been extensively investigated, and much less in a comparative analysis between subsets. Therefore, we examined the expression of GFP (A$_{2A}$R) within the tumor microenvironment. Having confirmed an important role for A$_{2A}$R in the control of AT3ova tumors (Fig. 2A), consistent with our previous observations[8], GFP expression was determined in the tumors and draining lymph nodes of mice bearing AT3ova tumors (representative flow cytometry Fig. 2B, gating strategy; Supplementary Fig. 1A). This analysis revealed that, within the draining lymph nodes, A$_{2A}$R expression was most abundantly expressed on NK cells on a per cell basis (mean 74% positive), followed by γδ T cells (49.3% positive) and type I NK T cells (33.7% positive) (Fig. 2C). Expression of A$_{2A}$R was lower on CD8$^+$ and CD4$^+$ lymphocytes (4.7% and 5.4% positive respectively) and virtually absent on B lymphocytes (1.9% positive). However, over 50% of the A$_{2A}$R positive cells within draining lymph nodes were either CD4$^+$ or CD8$^+$ lymphocytes because of their higher overall frequency relative to innate lymphocyte subsets (Fig. 2D). We next investigated the expression of A$_{2A}$R within tumor-infiltrating lymphocytes. Notably, A$_{2A}$R expression was highly expressed in all subsets except for B lymphocytes (Fig. 2E), with NK cells, CD4$^+$ lymphocytes and CD8$^+$ lymphocytes together constituting over 50% of A$_{2A}$R$^+$ cells within the tumor (Fig. 2F). Comparison of the expression of A$_{2A}$R in these lymphocyte subsets indicated that A$_{2A}$R was significantly upregulated in tumor-infiltrating lymphocytes relative to their counterparts in the tumor-draining lymph node, but this was more apparent with conventional CD8$^+$ and CD4$^+$ lymphocytes (5 fold increased expression in tumor relative to draining lymph node) than with NK, NKT and γδ T cells (all ~1.5 fold increased expression in the tumor) (Fig. 2B, G). Within the tumors, NK cells expressed a significantly higher level of A$_{2A}$R than all other immune subsets analyzed on a per cell basis (Fig. 2H). A similar pattern of A$_{2A}$R expression was observed in MC38 and E0771 tumors, that is to say A$_{2A}$R was highly expressed on innate-like cells (NK cells, NKT cells, γδ T cells) isolated from either DLNs or tumors and was significantly increased on conventional T lymphocytes isolated from tumors relative to counterparts isolated from draining

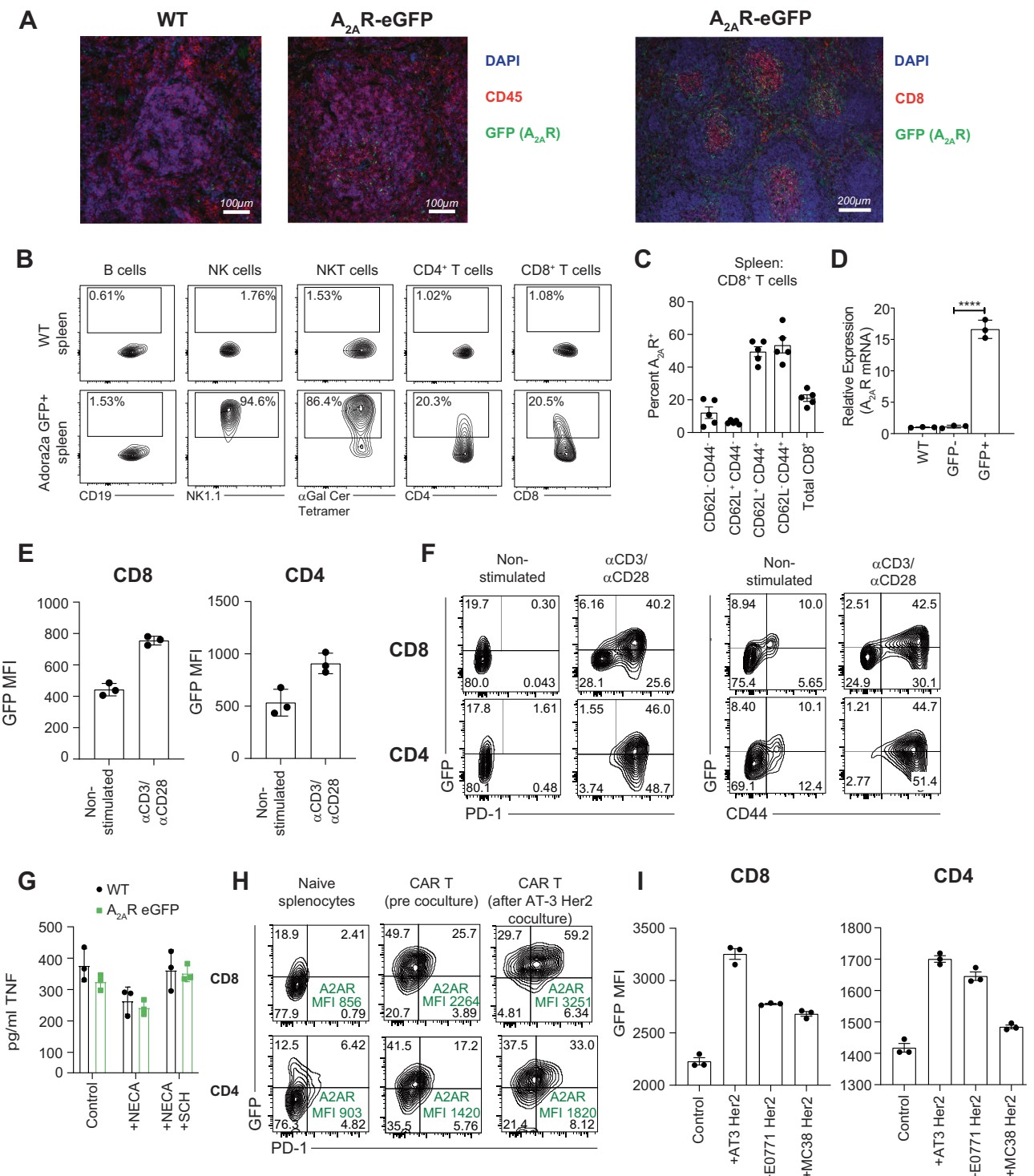

**Fig. 1 | Validation of A₂ₐR eGFP reporter mice. A** Immunofluorescence staining of sectioned spleens obtained from C57BL/6 A₂ₐR eGFP reporter or C57BL/6 WT mice (*n* = 1 experiment of 2 individual mice). Image shows A₂ₐR expression (GFP), CD45 (Red) or CD8 (Red) and DAPI (Blue) at 20x magnification (Left) or 10x magnification (Right). Section on right depicts GFP (A₂ₐR) expression on cells located within the arterioles. **B** A₂ₐR (GFP) expression on indicated splenic immune subsets from A₂ₐR eGFP or C57BL/6 WT mice as determined by flow cytometry. Representative experiment of *n* = 3 **C** Percentage ( ± SEM) of indicated CD8⁺ subsets expressing A₂ₐR in spleens of *n* = 5 naïve mice. **D** A₂ₐR mRNA expression in FACS sorted GFP⁺ and GFP⁻ cells isolated from A₂ₐR eGFP⁺ reporter mouse spleens. Data represent the mean ± SD of 3 samples obtained from independent mice. A₂ₐR expression is plotted relative to L32. Statistical significance determined using unpaired two-sided *t* test. ****p < 0.0001 **E**, **F**. Expression of GFP (A₂ₐR), PD-1 and CD44 in CD8⁺ and

CD4⁺ lymphocytes activated for 24 h with anti-CD3 (0.5 µg/ml) and anti-CD28 (0.5 µg/ml). Data represent the mean ± SD of triplicate samples from a representative experiment of *n* = 3. **G** Splenocytes from WT or A₂ₐR eGFP reporter mice were stimulated for 72 h with anti-CD3 (0.5 µg/ml) and anti-CD28 (0.5 µg/ml) in the presence or absence of NECA (1 µM) or SCH58261 (1 µM). Data represent the mean ± SD of triplicate samples. **H**, **I**. Anti-Her2 CAR T cells were generated from A₂ₐR eGFP mouse splenocytes and cocultured with indicated tumor cells. GFP Mean Fluorescence Intensity (MFI) of CD8⁺ T cells and CD4⁺ T cells (bottom) was measured by flow cytometry following 16 h co-culture with indicated Her2 expressing tumor lines, AT3 Her2, E0771 Her2 and MC38 Her2. **I** Data represent the mean ± SD of triplicate samples from a representative experiment of *n* = 3. Source data are provided as a Source Data file.

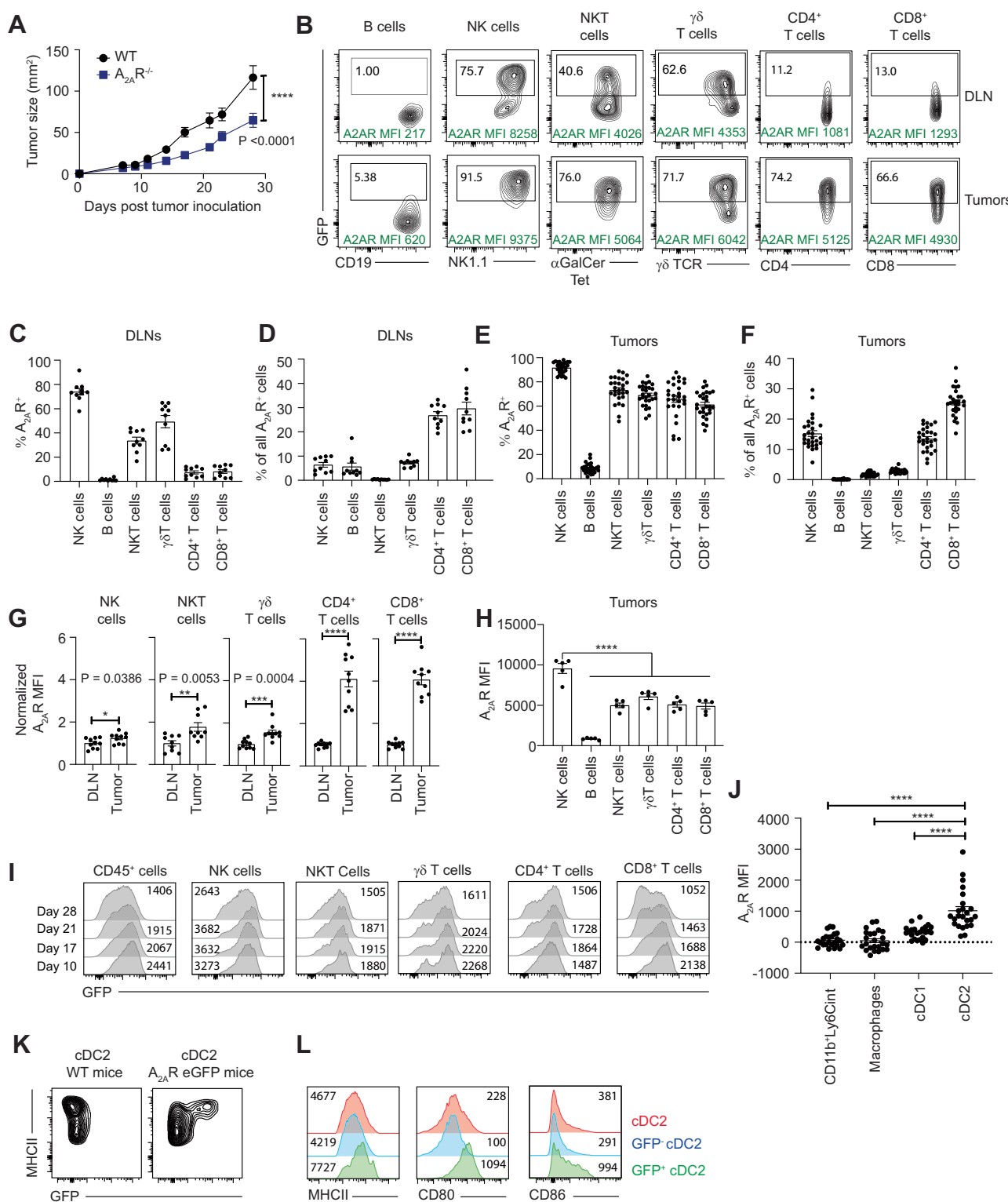

lymph nodes (Supplementary Fig. 1B-F, Supplementary Fig. 2A, B). To investigate the relationship between $A_{2A}R$ and tumor progression we assessed the expression of GFP on each of these subsets within AT3ova tumors at days 10, 17, 21 and 28 post tumor inoculation. Interestingly, $A_{2A}R$ expression was at its highest point within total CD45$^+$ cells at day 10 post-tumor inoculation and gradually decreased over time (Fig. 2I). This effect was largely caused by reduced $A_{2A}R$ expression within CD8$^+$ T cells, whilst the expression of $A_{2A}R$ in other subsets remained relatively stable. A similar reduction of $A_{2A}R$ expression in CD8$^+$ T cells over time was observed in E0771 tumors, albeit that the time course was

shorter than in the AT3ova model due to the more aggressive growth of this tumor line (Supplementary Fig. 2C). These data suggest that early intervention with $A_{2A}R$ antagonists may be critical for achieving maximum impact on CD8$^+$ T cell mediated anti-tumor immunity. Given previous reports indicating a role for $A_{2A}R$ expression on myeloid cells, we next assessed the expression of $A_{2A}R$ on myeloid subsets isolated from either AT3ova or E0771 tumors using a panel including antibodies for CD11b, CD11c, F4/80, Ly6C, CD103, XCR1, CD64, and MHCII. These analyses revealed that, amongst all myeloid subsets, $A_{2A}R$ expression was significantly higher on CD11b$^+$ cDC2 cells (defined as

**Fig. 2 | A$_{2A}$R expression is elevated on tumor-infiltrating T cells and cDC2s.**
**A** WT, A$_{2A}$R$^{-/-}$ or A$_{2A}$R eGFP reporter mice were inoculated sub-cutaneously with
$5 \times 10^5$ AT-3 ova tumors. **A.** Tumor size was measured over time (mm$^2$). Data
represent the mean ± SEM of 6 (WT) or 7 (A$_{2A}$R$^{-/-}$) mice per group. **B–L** Immune cells
from draining lymph nodes and tumors were analyzed by flow cytometry at day 21
post-tumor inoculation unless otherwise stated. **B** Representative flow cytometry
plots from concatenated samples. **C, E** Percentage of indicated cell types express-
ing GFP (A$_{2A}$R) within DLNs (C) or tumors (E). **D, F** Proportion of all GFP$^+$ cells that
are of indicated lineage within DLNs (D) or tumors (F). **C, D** Data represent the
mean ± SEM of 10 mice per group from 2 pooled experiments. **E, F** Data represent
the mean ± SEM of 28 mice per group from 5 pooled experiments. **G** Relative
expression of GFP (A$_{2A}$R) for indicated cell types in the DLN and tumor. MFI was
normalized such that expression in the DLN was equal to 1. Data represent the
mean ± SEM of 10 mice per group from 2 pooled experiments. *$p < 0.05$, **$p < 0.01$,
***$p < 0.001$, ****$p < 0.0001$, unpaired two-sided $t$ test. **H** MFI of GFP (A$_{2A}$R)
expression by each cell type isolated from AT3ova tumors. Data represented as
mean ± SEM of 5 mice per group. ****$p < 0.0001$, statistics determined by one-way
ANOVA **I** Expression of GFP (A$_{2A}$R) in indicated cell types isolated from tumors at
indicated timepoints. Numbers represent the GFP MFI. Data represent con-
catenated samples from $n = 6$ mice. **J** Expression of GFP (A$_{2A}$R) in indicated myeloid
populations. ΔGFP expression calculated as GFP MFI minus the background fluor-
escence for each cell type as observed in WT mice. Data represented as the
mean ± SEM from 24 mice per group, pooled from 4 independent experiments.
****$p < 0.0001$, statistics determined by one way ANOVA. **K** Expression of MHCII and
GFP on cDC2 cells (Gated as CD45$^+$TCRβ$^-$NK1.1$^-$Ly6C$^-$CD64$^-$CD11c$^+$MHCII$^+$F4/
80$^{low}$CD103$^-$XCR1$^-$). **L** Expression of MHCII, CD80, and CD86 on total cDC2 (Red),
GFP$^-$ cDC2 (Blue) and GFP$^+$ cDC2 (Green). **K, L.** Data concatenated from 6 individual
mice. Source data are provided as a Source Data file.

Ly6c$^-$CD64$^-$CD11c$^+$MHCII$^+$CD103$^-$XCR1$^-$F4/80$^{low/dim}$), relative to all other
myeloid cell subsets (Fig. 2J, Supplementary Fig. 2D). Interestingly the
GFP$^+$ subset of cDC2s exhibited significantly higher expression of
MHCII, CD80 and CD86 (Fig. 2K, L, Supplementary Fig. 2D), indicating
that this population may represent a more mature subset of cDC2 cells
with enhanced capacity to prime T cell responses.

## A$_{2A}$R expression is indicative of CD8$^+$ T cell activation status within draining lymph nodes but less so within tumors

Given that A$_{2A}$R expression was significantly enhanced on tumor-
infiltrating CD8$^+$ and CD4$^+$ lymphocytes within the tumor micro-
environment, we further analyzed the phenotype of A$_{2A}$R$^+$ T cells
relative to their A$_{2A}$R$^-$ counterparts. This analysis included the
expression of markers associated with differentiation status (CD62L,
SLAMF6, CD69, CD44), the immune checkpoint PD-1 and CD39, the
ectoenzyme responsible for the breakdown of ATP to AMP. Notably,
CD39 is also a recognized marker of tumor-reactive terminally differ-
entiated CD8$^+$ T cells and its expression limits anti-tumor immunity[31–33].
A SIINFEKL-loaded tetramer reagent was also used to identify tumor-
antigen (Ova) specific CD8$^+$ T cells. We first investigated the expression
of A$_{2A}$R on tetramer$^+$ and tetramer$^-$ subsets. Within tumor-draining
lymph nodes A$_{2A}$R expression was significantly upregulated on
CD44$^+$tetramer$^+$ CD8$^+$ T cells relative to other CD8$^+$ T cell subsets
(Fig. 3A), consistent with the notion that A$_{2A}$R was upregulated fol-
lowing activation of these cells by tumor antigens presented by APCs in
the draining lymph nodes. However, within the tumors there was only
a modest increase in A$_{2A}$R expression on CD62L$^-$ tetramer positive cells
relative to CD62L$^-$ tetramer negative counterparts (Fig. 3B, Supple-
mentary Fig. 1G). Within tumors, CD8$^+$ T cells follow a pathway of
differentiation from TCF7$^+$ precursor exhausted cells to TCF7$^-$ term-
inally differentiated cells. Given the technical complexities of
TCF7 staining in GFP reporter mice due to quenching of GFP following
nuclear permeabilization, we employed the gating strategy described
by Beltra et al. using the markers CD69 and SLAMF6 whereby T cells
transition from CD69$^+$SLAMF6$^+$ > CD69$^-$SLAMF6$^+$ > CD69$^-$SLAMF6-
>CD69$^+$, SLAMF6$^{-34}$. Analysis of these distinct subsets indicated that
A$_{2A}$R was modestly but significantly enriched within the
CD69$^+$SLAMF6$^+$ subset relative to SLAMF6- counterparts (Fig. 3C).
Similarly, within E0771 tumor cells it was observed that A$_{2A}$R expres-
sion was significantly higher on SLAMF6$^+$CD39$^-$ or SLAMF6$^-$CD39$^+$ cells
relative to SLAMF6$^-$CD39$^-$ counterparts (Supplementary Fig. 2E). We
next investigated whether these differences contributed to the
decreased expression of A$_{2A}$R observed in CD8$^+$ tumor infiltrating
lymphocytes over time (Fig. 2I). We observed that in the course of
tumor progression that the proportion of both the antigen specific
tetramer positive population and the SLAMF6$^+$CD69$^+$ progenitor
population were progressively diminished (Supplementary Fig. 1H).
Given that these cell types were the highest expressers of A$_{2A}$R this
partly accounts for the reduced expression of A$_{2A}$R in total CD8$^+$ T cells
over time. Moreover, on a per cell basis these cell populations elicited

significantly higher expression of A$_{2A}$R at day 10 than other timepoints
but there was no significant difference from day 17 onwards (Supple-
mentary Fig. 1H). Therefore, the reduction in A$_{2A}$R expression in CD8$^+$
T cells over time is partly explained by a reduced frequency of the
tetramer positive and SLAMF6$^+$CD69$^+$ cells and partly due to reduced
expression of A$_{2A}$R by these subsets on a per cell basis at later time-
points. To confirm this using a complementary approach we deter-
mined the expression of A$_{2A}$R following treatment of mice with
FTY720. FTY720 acts to block S1PR1 and thus prevent the egress of
immune cells from lymph nodes into the tumor site. We reasoned that
if A$_{2A}$R was expressed more highly on terminally differentiated cells
that GFP expression would be increased following FTY720 treatment
since this treatment would prevent cells egressing from the draining
lymph node and replenishing the SLAMF6$^+$CD69$^+$ population. FTY720
treatment was effective, as demonstrated by significantly reduced
numbers of CD8$^+$ and CD4$^+$ T cells in the peripheral blood (Supple-
mentary Fig. 1I). Within tumors, FTY720 treatment resulted in a
reduction in the proportion of CD69$^+$SLAMF6$^+$ (less-differentiated
population) CD8$^+$ T cells and A$_{2A}$R expression was significantly
reduced (Fig. 3D), supporting our earlier observations that A$_{2A}$R
expression was not enhanced on more terminally differentiated CD8$^+$
T cells.

To further investigate this relationship, we determined the
expression of A$_{2A}$R on PD-1$^+$ and CD39$^+$ subsets. Somewhat surpris-
ingly and counter to our observations in vitro, A$_{2A}$R was expressed
equally between PD-1$^+$/ PD-1$^-$ and CD39$^+$/CD39$^-$ subsets, whereas PD-1 and
CD39 were largely co-expressed. Thus, A$_{2A}$R was not co-expressed with
other known inhibitory checkpoint receptors on CD8$^+$ T cells sug-
gesting that A$_{2A}$R blockade may modulate the function of distinct
subsets of CD8$^+$ T cells (Fig. 3E).

In contrast to CD8$^+$ T cells, both CD39 and PD-1 were expressed to a
significantly greater extent on CD4$^+$A$_{2A}$R$^+$ cells relative to their A$_{2A}$R$^-$
counterparts (Fig. 3F). This may reflect an increased expression of A$_{2A}$R
on Treg cells, which are known to express high levels of CD39 and PD-1
in the tumor microenvironment[35–38], particularly given CD4$^+$GFP$^+$CD39$^+$
cells were enriched for CD25$^+$ cells relative to CD4$^+$GFP$^+$CD39$^-$ coun-
terparts (Fig. 3G). However, this could not be confirmed with Foxp3
counterstaining due to the issues of GFP degradation upon nuclear
permeabilization as aforementioned. Taken together, these results
suggest that TCR-mediated activation/ differentiation may not be the
major factor driving A$_{2A}$R expression within the tumor microenviron-
ment on CD8$^+$ T cells.

## The efficacy of immune checkpoint blockade is correlated with the emergence of PD-1$^+$A$_{2A}$R$^-$ cells following therapy

Given previous studies indicating that A$_{2A}$R blockade enhances the
efficacy of conventional immune checkpoint blockade or
chemotherapy[5,8,14] and the clinical interest in targeting this pathway,
we next assessed the effect of anti-PD-L1, anti-CTLA-4, the combination
of anti-PD-L1 and anti-CTLA-4 or carboplatin on the expression of the

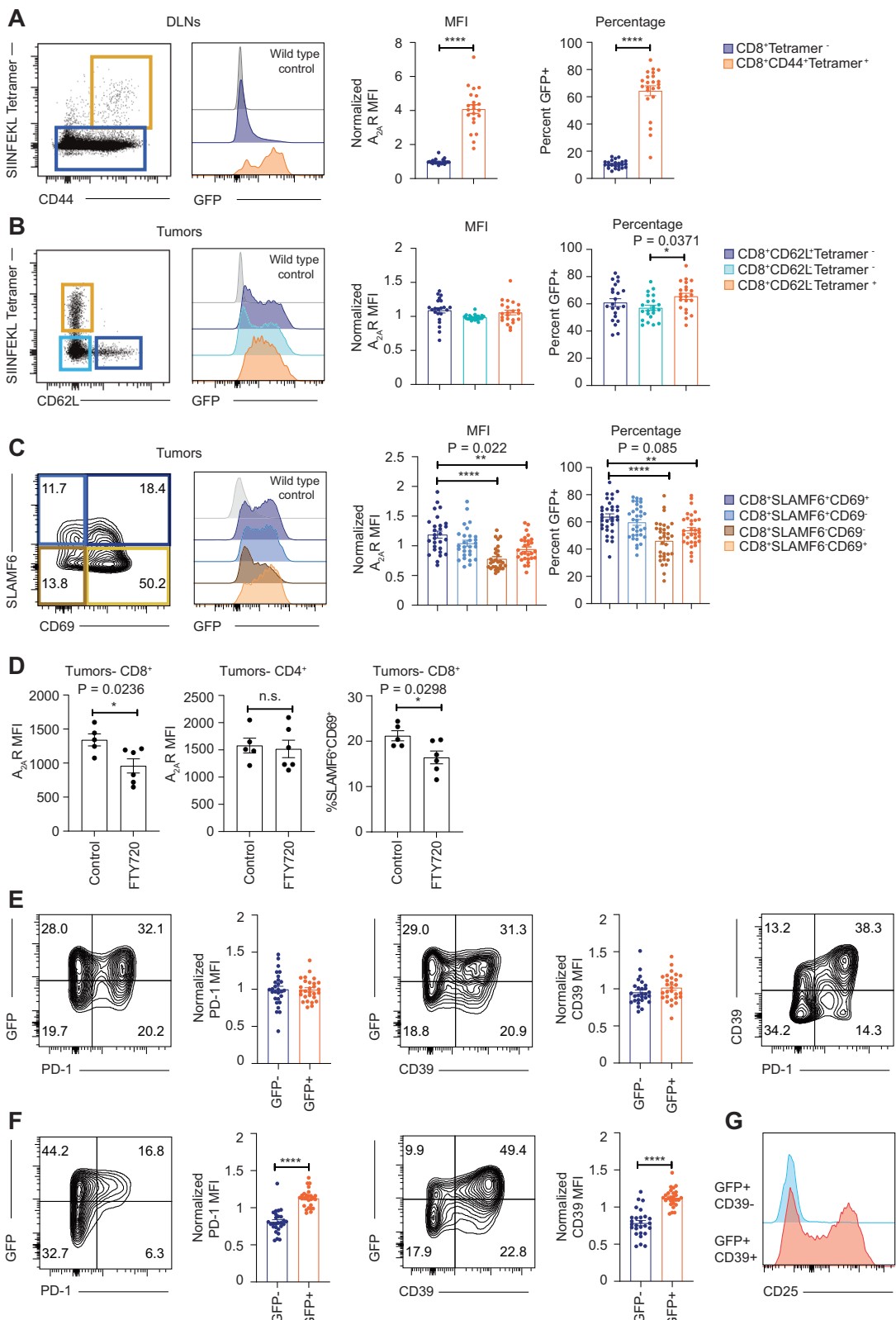

A₂AR within the tumor microenvironment. As expected, immune checkpoint blockade therapy resulted in significantly reduced tumor weights and an increase in tumor-infiltrating CD8⁺ lymphocytes (Supplementary Fig. 3A, B). Immunofluorescent analyses confirmed that combination (anti-PD-L1 and anti-CTLA-4) therapy resulted in increased numbers of CD45⁺ cells infiltrating the tumors, and this was associated with an increased abundance of GFP⁺ cells (Fig. 4A).

In terms of T lymphocyte phenotype, whilst the overall frequency of CD8⁺A₂AR⁺ (but not CD4⁺A₂AR⁺) T cells was increased by therapy due to an increase in the proportion of CD8⁺ T cells as determined by flow cytometry analyses (Fig. 4B), the expression of A₂AR on a per cell basis was not significantly modulated by these treatment regimens at day 7 post-therapy (Fig. 4C). Moreover, carboplatin treatment also failed to modulate A₂AR expression on CD8⁺ or CD4⁺ lymphocytes

**Fig. 3 | A₂AR expression is elevated on tumor antigen-specific CD8⁺ T cells within draining lymph nodes but not tumors.** A–G A₂AR eGFP reporter mice were injected subcutaneously with $5 \times 10^5$ AT-3 ova tumors. Draining lymph nodes (DLN; A) and tumor infiltrating lymphocytes (B–G) were analyzed by flow cytometry at day 21 post-tumor inoculation. Flow Cytometry plots represent data from concatenated samples in one representative experiment. A Expression of GFP (A₂AR) in CD8⁺ T cells that are CD44⁺Tetramer⁺ or Tetramer⁻. Data represent the mean ± SEM of 22 individual mice pooled from 4 experiments. B, C Expression of GFP (A₂AR) on indicated subsets of tumor-infiltrating CD8⁺ T cells. Data represent the mean ± SEM of 17 mice pooled from 3 individual experiments (B) or from 28 mice per group pooled from 5 individual experiments (C). D Mice were treated at days 14, 16, 18, and 20 post-tumor inoculation with 25 µg FTY720. Data represent the mean ± SEM of 5 (Control) or 6 (FTY720) mice per group. E, F Coexpression of A₂AR, PD-1 and CD39 on CD8⁺ T cells (E) and CD4⁺ T cells (F). Data represent the mean ± SEM of 28 (E) or 27 (F) mice per group pooled from 5 independent experiments. G Expression of CD25 on indicated CD4⁺ subsets. Data are concatenated from 6 individual samples. Where normalized MFI is shown, data were normalized relative to the average MFI of total CD8⁺ T cells in each experiment. *$p < 0.05$, **$p < 0.01$, ***$p < 0.001$, ****$p < 0.0001$. Statistics determined by one-way ANOVA (B, C) or unpaired two-sided *t* test (A, D, E, F). Source data are provided as a Source Data file.

(Supplementary Fig. 3C). To further interrogate the impact of immune checkpoint blockade on A₂AR expression, and to investigate the possibility that A₂AR was transiently upregulated following treatment, further experiments were performed to determine the expression of GFP on CD8⁺ tumor-infiltrating lymphocytes (TILs) at days 2 and days 4 post-treatment with either anti-PD-L1 or anti-PD-L1 and anti-CTLA-4. Consistent with our analyses at day 7, no significant modulation of A₂AR expression was observed (Supplementary Fig. 3D). Although immune checkpoint blockade did not modulate A₂AR expression on a per cell basis, the expression of PD-1 was increased on CD8⁺ T cells as expected (Supplementary Fig. 3E, F) and further analysis of the activated PD-1⁺ cells that emerged following therapy revealed that anti-PD-L1 or anti-PD-L1 and anti-CTLA-4 treatment induced a significant increase of PD-1⁺A₂AR⁻ cells (Fig. 4D). Similarly, a significant increase in CD8⁺PD-1⁺A₂AR⁻ cells (but not PD-1⁺A₂AR⁺ cells) were observed in MC38 tumors following treatment with anti-PD-L1 (Supplementary Fig. 3G). Interestingly, the emergence of these cells was positively correlated with increased therapeutic efficacy in response to anti-PD-L1 single agent therapy (Fig. 4E, F). Lastly, the expression of A₂AR in draining lymph nodes was assessed in the context of therapy. In contrast to the lack of modulation of A₂AR expression within the tumor by anti-PD-L1 therapy, a significant increase in the proportion and absolute number of tetramer⁺A₂AR⁺ CD8⁺ T cells within the draining lymph nodes was observed, suggesting that these cells may be more susceptible to adenosine mediated immunosuppression post-treatment (Fig. 4G).

Finally, given that we observed the expression of A₂AR on myeloid subsets including cDC2s within the tumor microenvironment (Fig. 2J–L), we also assessed the impact of immunotherapy on the expression of A₂AR on myeloid cells. Interestingly, anti-CTLA-4 significantly enhanced the expression of A₂AR on cDC2 and, to a lesser extent, cDC1s, when combined with anti-PD-L1 (Supplementary Fig. 4A, B). CTLA-4 blockade has previously been shown to induce upregulation of CD80 and CD86[39] and thus we hypothesized that A₂AR expression may be indicative of an activated cDC2 population. To investigate this, we generated dendritic cells from bone marrow of A₂AR eGFP reporter mice. Consistent with our observations in tumors, a subset of cDC2 were identified to express A₂AR that also exhibited increased expression of CD80 and CD86 (Supplementary Fig. 4C, D). Furthermore, activation of bone marrow derived dendritic cells with poly IC resulted in cDC2 maturation, as evidenced by increased expression of MHCII, CD80 and CD86, which was concomitant with enhanced A₂AR expression (Supplementary Fig. 4C, D). To determine the consequence of adenosine signaling on bone marrow-derived dendritic cell (BMDC) function, they were cocultured with NECA, a pan adenosine receptor agonist, prior to coculture with naïve T cells in a T cell priming assay. These experiments revealed that adenosine signaling resulted in significantly attenuated ability of BMDCs to induce T cell proliferation (Supplementary Fig. 4E).

## A₂AR⁺ CD8⁺ TILs exhibit reduced STAT5 transcriptional signatures

To further investigate the phenotype of A₂AR⁺ CD8⁺ T cells within tumors, we performed 3'RNA-sequencing on CD8⁺GFP⁺ and CD8⁺GFP⁻ tumor-infiltrating T cells both at baseline and in the context of anti-PD-

L1 treatment. As expected GFP⁻ and GFP⁺ cells clustered differently as did anti-PD-L1 and non-treated samples (Fig. 5A). Analysis of differentially expressed genes between GFP⁺ and GFP⁻ cells confirmed that A₂AR was more highly expressed in CD8⁺GFP⁺ cells (Fig. 5B). Pathway analysis of differentially expressed genes revealed significant negative enrichments for "Interferon alpha response", "Interferon gamma response" and for the expression of STAT5 target genes based on a signature developed by Grange et al.[40] within A₂AR⁺ cells (Fig. 5C, D, Supplementary Fig. 5). This gene signature included notable effector genes such as *Ifng, Tnf, Prf1* and *Gzmb*. The association between A₂AR⁺ status and reduced STAT5 target genes was of interest because of our previous observations that A₂AR agonists negatively regulate JAK-STAT signaling[16] and a previous report from Cekic and colleagues that IL7R signaling protects CD8⁺ T cells from adenosine mediated-suppression[41]. Although Cekic et al. attributed the impact of IL7 signaling on adenosine-mediated suppression to the inactivation of FOXO1, the activation of STAT5 signaling by IL7 represents a possible complementary mechanism of action. Interestingly, treatment with anti-PD-L1 significantly enhanced STAT5 target genes in both A₂AR⁺ and A₂AR⁻ cells, such that there was no longer a significant difference in the expression of these genes between the two subsets (Fig. 5E). This infers that one consequence of anti-PD-L1 treatment is to overcome reduced STAT5 signaling in A₂AR⁺ CD8⁺ T cells.

## A₂AR is potently upregulated by IL-18 and limits the anti-tumor efficacy of IL-18

We next investigated the capacity of cytokines to upregulate A₂AR expression. Beginning with a broad screen of 23 cytokines on the expression of A₂AR on CD8⁺ T lymphocytes derived from splenocytes, we identified that amongst the most potent inducers of A₂AR expression were members of the IL-1 family of cytokines including IL-1β, IL-18 and IL-36 (Fig. 6A). We proceeded to investigate the impact of the top hits from this in vitro screen on the expression of A₂AR in tumor-infiltrating lymphocytes ex vivo. These results revealed that IL-12 and IL-18 increased the expression of A₂AR on multiple immune lineages including CD8⁺ T cells and NK cells (Fig. 6B, Supplementary Fig. 6A). Given these data and the clinical interest in IL-18 based therapies[42,43], we further investigated the relationship between IL-18 and A₂AR. In repeat experiments we confirmed that IL-18 was a potent inducer of A₂AR expression in both murine T cells (Fig. 6C) and human NK cells (Fig. 6D). To investigate this further we engineered AT3ova and MC38 tumor cells to express IL-18. Despite successful transduction of both cell lines, only MC38 secreted detectable levels of IL-18 (Supplementary Fig. 6B). We therefore evaluated the growth of MC38-IL-18 or MC38 mCherry control tumors and observed a significant reduction in growth of IL-18 expressing tumors (Supplementary Fig. 6C). To investigate the relationship between IL-18 and A₂AR expression in this context we inoculated A₂AR eGFP reporter mice with either IL-18 expressing or control MC38 tumors and assessed the phenotype of tumor-infiltrating lymphocytes. This analysis revealed that CD8⁺ T cells, CD4⁺ T cells and NK cells derived from IL-18 expressing tumors exhibited significantly increased levels of A₂AR expression, whereas we observed no significant increase in PD-1 expression on CD8⁺ T cells (Fig. 6E, F). To evaluate whether A₂AR expression limited the efficacy of

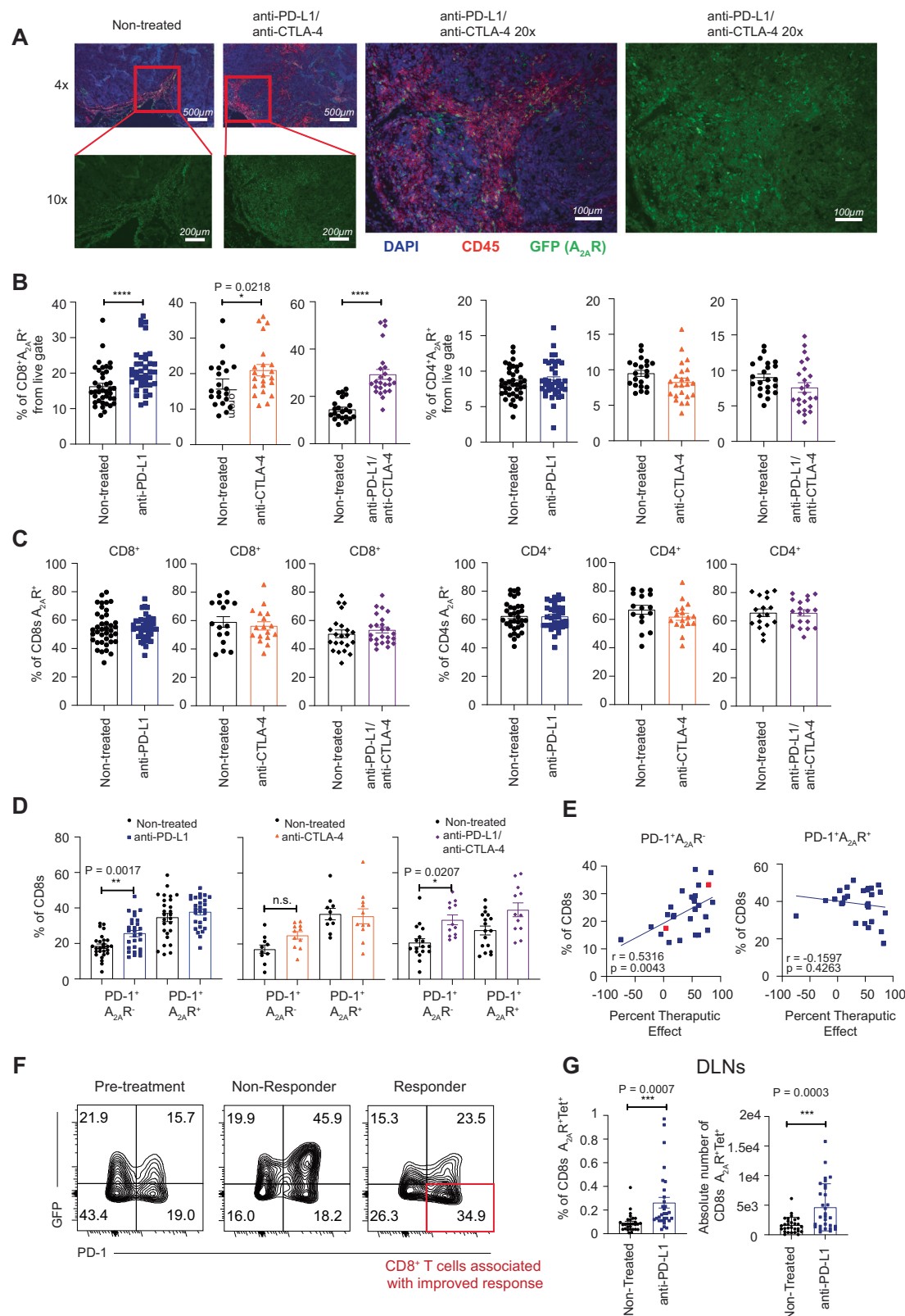

IL-18 therapy we inoculated control or IL-18 expressing MC38 cells into $A_{2A}R$ deficient or wild-type mice. Strikingly, $A_{2A}R$ deficient mice elicited more pronounced anti-tumor responses with significantly reduced tumor growth and increased survival relative to WT counterparts (Fig. 6G, H; Supplementary Fig. 6D). Depletion experiments revealed that the anti-tumor efficacy of IL-18 was dependent on both $CD8^+$ T cells and NK cells (Fig. 6I). Lastly, to evaluate the interplay

between IL-18 and $A_{2A}R$ expression in a more clinically relevant system we determined the impact of recombinant IL-18 treatment on the expression of $A_{2A}R$ within the context of established MC38 tumors. These experiments revealed that $A_{2A}R$ expression was significantly increased on $CD8^+$ T cells and NK cells following treatment with IL-18 (Fig. 6J). These data highlight the potential of combining IL-18 therapy with $A_{2A}R$ blocking therapies such as small molecule antagonists and

**Fig. 4 | PD-L1 blockade results in the emergence of a PD-1⁺A₂ₐR⁻ subset of CD8⁺ T cells that correlates with therapeutic outcomes. A–G.** A₂ₐR eGFP reporter mice were injected with $5 \times 10^5$ AT-3 ova tumors subcutaneously and where indicated treated at days 14 and 18 post tumor inoculation with anti-PD-L1 (200 µg/ mouse) and anti-CTLA-4 (150 µg/ mouse). **A** Tumors were excised 21 days post-tumor inoculation and fixed with 4% paraformaldehyde followed by 30% sucrose overnight. Immunofluorescence staining was performed on tumor slides. Staining for CD45 (red), DAPI (blue) and A₂ₐR (green) is shown. Slides imaged using ×4 and ×10 objective lens. Representative image from one mouse is shown. **B** Percentage of live, CD45.2⁺ cells that were A₂ₐR⁺ CD8⁺ T cells (left) and A₂ₐR⁺ CD4⁺ T cells (right). Data represent the mean ± SEM of individual mice pooled from 3–7 experiments (control and anti-PD-L1 $n = 39$, 7 experiments, anti-CTLA-4 $n = 23$, 4 experiments, anti-PD-L1 and anti-CTLA-4 $n = 18$, 3 experiments). **C** Percentage of CD8⁺ T cells (left) or CD4⁺ T cells (right) that are A₂ₐR⁺ following treatment. Data represent the mean ± SEM of individual mice pooled from 4 to 7 experiments (CD8⁺: control $n = 39$, 7 experiments, anti-PD-L1 $n = 40$, 7 experiments, anti-CTLA-4 $n = 17$, 3 experiments anti-PD-L1 and anti-CTLA4 $n = 24$, 4 experiments, CD4⁺: control $n = 34$, and anti-PD-L1 $n = 34$, anti-CTLA-4 $n = 17$, anti-PD-L1 and anti-CTLA4 $n = 18$)

**D** Percentage of CD8⁺ T cells exhibiting a PD-1⁺A₂ₐR⁻ or PD-1⁺A₂ₐR⁺ phenotype at day 7 post treatment. Data represent the mean ± SEM of individual mice pooled from 2 to 6 experiments (control $n = 33$, 6 experiments, anti-PD-L1 $n = 28$, 5 experiments anti-CTLA-4 $n = 11$, 2 experiments, anti-PD-L1 and anti-CTLA4 $n = 12$, 2 experiments). **E** Correlation of the frequency of CD8⁺ T cells exhibiting PD-1⁺A₂ₐR⁻ or PD-1⁺A₂ₐR⁺ phenotypes with therapeutic efficacy. Therapeutic efficacy calculated as 100-(tumor weight in test sample/ average tumor weight in control group*100). $n = 27$ mice. **F** Representative flow cytometry staining of CD8⁺ T cells derived from non-treated mice (left) and mice treated with anti-PD-L1 that either elicited a therapeutic response (right) or no response (center). Data points represented as non-responders and responders are depicted as red in (**E**). **G** Proportion of CD8⁺ T cells isolated from tumor-draining lymph nodes exhibiting an A₂ₐR⁺tetramer⁺ phenotype (left) or absolute numbers of these cells (right). Data represent the mean ± SEM of 29 (non-treated) or 30 (anti-PD-L1) mice pooled from 5 individual experiments. *$p < 0.05$, **$p < 0.01$, ***$p < 0.001$, ****$p < 0.0001$, unpaired two-sided $t$ test (**B**, **D**, **G**) or Linear regression analysis (**E**). Source data are provided as a Source Data file.

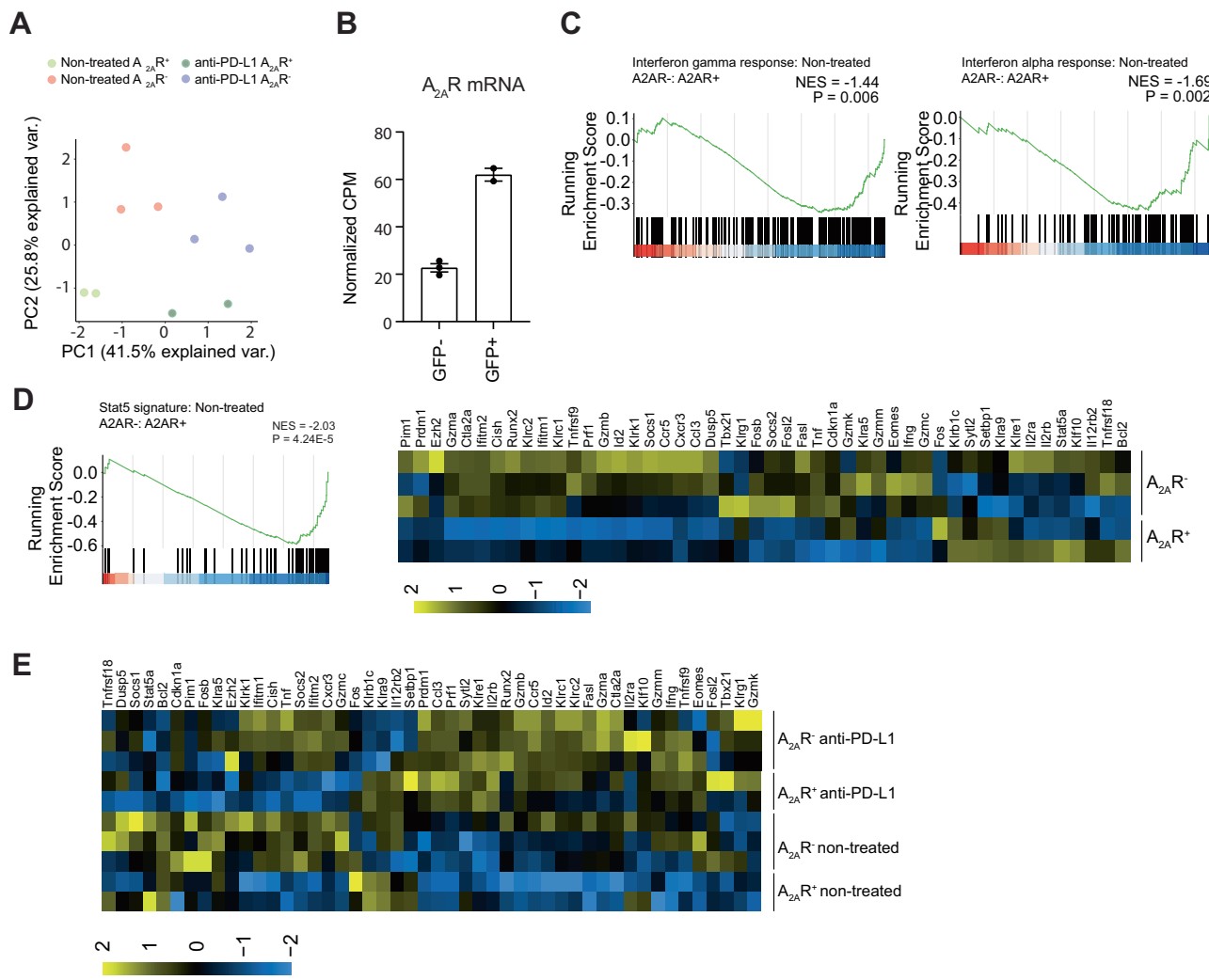

**Fig. 5 | A₂ₐR positive CD8⁺ T cells exhibit decreased expression of STAT5 target genes prior to treatment with anti-PD-L1.** A₂ₐR eGFP reporter mice were injected with $5 \times 10^5$ AT-3 ova tumors subcutaneously and were indicated treated at days 14 and 18 post-tumor inoculation with anti-PD-L1. At day 21 post tumor inoculation CD8⁺ tumor-infiltrating lymphocytes were analyzed by 3'RNA-sequencing. Biological samples were pooled from $n = 3$ mice per replicate. Data represented as mean ± SD. **A** Principle component analysis based on the top 100 most variable genes. **B** Normalized CPMs for *Adora2a* expression from CD8⁺GFP⁻ ($n = 3$) and CD8⁺GFP⁺ ($n = 2$) cells. **C–E** Gene set enrichment analysis or heatmap of gene expression for indicated pathways comparing CD8⁺A₂ₐR⁺ and CD8⁺A₂ₐR⁻ cells. Raw P values are shown. Source data are provided as a Source Data file.

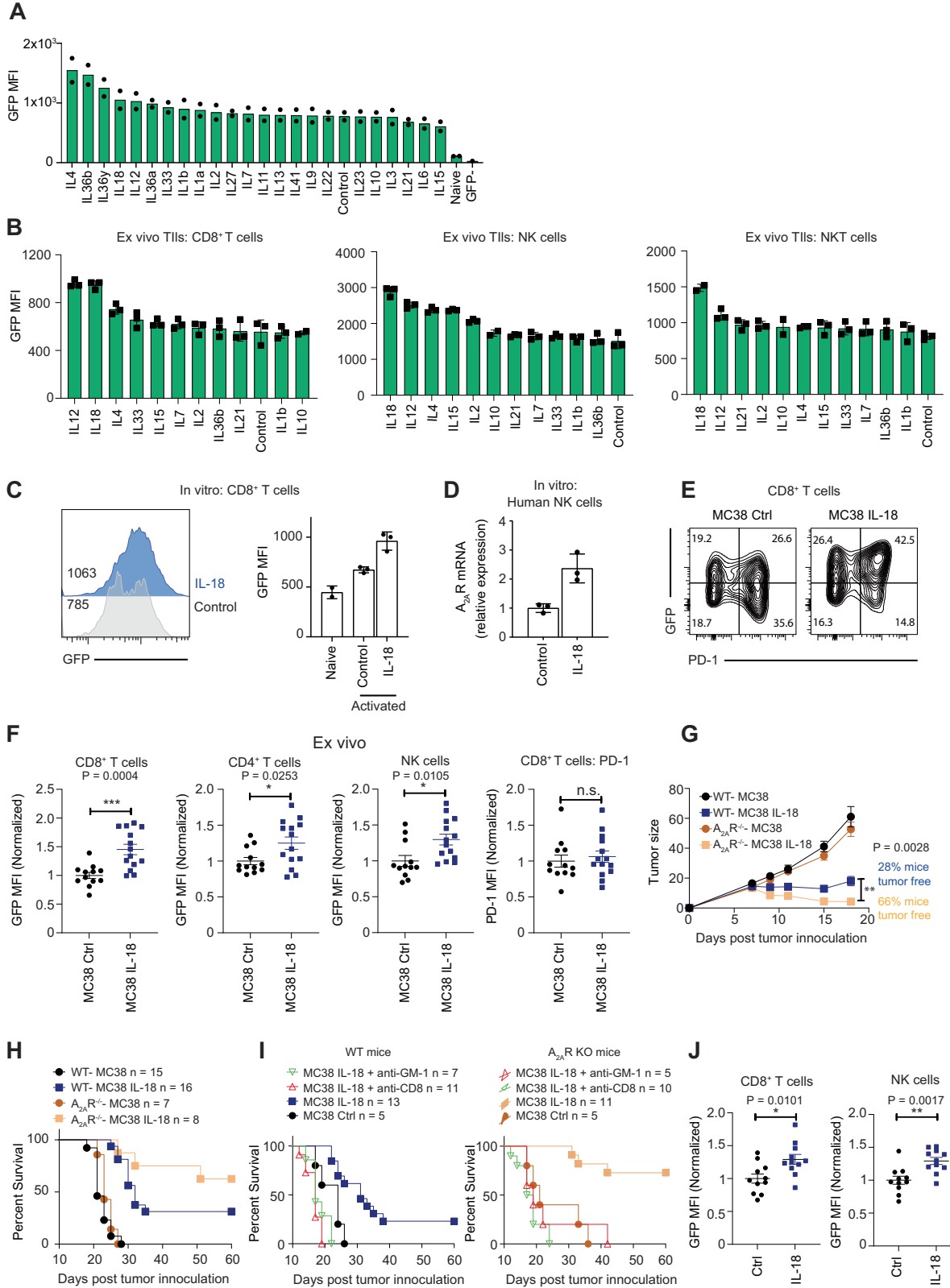

more broadly highlight the utility of this model for establishing rationale combination therapies with therapeutics designed to target the adenosine pathway.

## Discussion

Adenosine-mediated immunosuppression is a significant barrier to the formation of effective anti-tumor immune responses. Antibodies targeting the ectoenzymes responsible for the generation of adenosine, CD73 and CD39, and small molecule antagonists of the adenosine receptors $A_{2A}R$ and $A_{2B}R$, have been developed for clinical application[1,44,45]. $A_{2A}R$ can suppress several immune cell types; however, the pattern of expression of $A_{2A}R$ within the context of an anti-tumor immune response has remained largely unknown due to a lack of robust flow cytometry antibodies. We therefore sought to develop a

**Fig. 6 | IL-18 treatment leads to the upregulation of $A_{2A}R$ which consequently limits anti-tumor immunity. A** CD8$^+$ T cells were isolated from C57BL/6 $A_{2A}R$ eGFP or C57BL/6 WT mice and stimulated with plate-bound anti-CD3 and soluble anti-CD28 (2 μg/ml) mAbs plus indicated cytokines (100 IU/ml IL-2, or 50 ng/ml other cytokines) for 72hrs. Data represent the mean from 2 individual technical replicates. **B** AT3 ova tumor-infiltrating lymphocytes were isolated and cultured overnight with indicated cytokines (50 ng/ml). Data represent the mean ± SD of 3 technical replicates from a representative experiment of $n = 3$. **C** $A_{2A}R$ GFP splenocytes were treated for 3 days with anti-CD3/ anti-CD28 and, where indicated, IL-18 (10 ng/ml). Histogram overlay of concatenated samples (left) or mean ± SD of 2 (Naïve) or 3 (activated) replicate samples from a representative experiment of $n = 3$ (right). **D** Human NK cells were stimulated with 20 IU/ml IL-2 (control) or IL-2 and IL-18 (50 ng/ml) for 16 h. Expression of $A_{2A}R$ mRNA relative to GAPDH housekeeping gene. Data represent the relative expression (±SD) of triplicate samples. **E–J** C57BL/6 $A_{2A}R$ eGFP mice (**E, F**), C57BL/6 WT mice or $A_{2A}R^{-/-}$ mice (**G, H**) were injected with $1 \times 10^6$ MC38 mCherry or MC38 mCherry-IL-18 expressing tumor cells. **E, F** Expression of PD-1 and $A_{2A}R$ (GFP) by indicated subsets. Flow cytometry plots of concatenated samples from a representative experiment. **F** Data represented as the mean ± SEM of 12 (MC38 Ctrl) or 14 (MC38 IL-18) mice per group pooled from 2 experiments. Data were normalized relative to the average MFI of the relevant cell population in control tumors within each experiment. **G** Tumor growth represented as the mean ± SEM pooled from 2 representative experiments ($n = 15$ WT MC38, 17 WT MC38-IL-18, 10 $A_{2A}R$ KO MC38, 9 $A_{2A}R$ KO MC38-IL-18) **H** Survival of mice with 100mm$^2$ tumor size used to designate survival. ($n = 15$ WT control, 16 WT IL-18, 7 $A_{2A}R$ KO control, 8 $A_{2A}R$ KO IL-18). **I** C57BL/6 WT mice or $A_{2A}R^{-/-}$ mice were injected with $1.5 \times 10^6$ tumor cells and where indicated treated with anti-CD8 or anti-GM-1, $n = 5–13$ per group. **J** At day 12 post-tumor inoculation 1 μg IL-18 was administered intra-tumorally at 17 and 2 h prior to analysis of GFP expression. Data are represented as the mean ± SEM of 11 mice per group. *$p < 0.05$, **$p < 0.01$, ***$p < 0.001$ unpaired two-sided $t$ test (**F, J**), Two-way ANOVA (**G**) or log-rank (**H**) statistical test. Source data are provided as a Source Data file.

transgenic model to investigate the pattern of expression of $A_{2A}R$ in the context of anti-tumor immunity and following therapeutic intervention with either immunotherapy, chemotherapy, or cytokines.

We initially confirmed the veracity of the $A_{2A}R$ transgenic model through analysis of $A_{2A}R$ mRNA expression in GFP$^+$ and GFP$^-$ cells. Moreover, in line with our expectations, analysis of splenocytes from these mice revealed that GFP ($A_{2A}R$) was abundantly expressed in cell types known to express high levels of $A_{2A}R$ such as NK cells[24] and NKT cells[26] but was not expressed in B cells. Furthermore, GFP was strongly upregulated following T cell activation through the TCR or a CAR, in line with previous observations with $A_{2A}R$ mRNA. We therefore proceeded to investigate the expression of $A_{2A}R$ in the context of tumor-bearing mice. Consistent with our observations using splenocytes, $A_{2A}R$ was highly expressed on all innate-like subsets evaluated including NK cells, NKT cells and γδ T cells. Tumor-infiltrating CD8$^+$ and CD4$^+$ T cells also expressed high levels of $A_{2A}R$ and notably this expression was significantly higher than observed with CD8$^+$ or CD4$^+$ T cells isolated from spleen or draining lymph nodes. We originally hypothesized that this may relate to the activation status of the T cells given that TCR activation is known to drive expression of $A_{2A}R$ and indeed this is what we observed using the $A_{2A}R$ eGFP reporter mice in vitro. Whilst antigen (ova)-specific CD8$^+$ T cells expressed significantly higher levels of $A_{2A}R$ than non-ova-specific CD8$^+$ T cells in draining lymph nodes, this difference was more modest within tumors. Moreover, within tumors $A_{2A}R$ expression was similar between PD-1$^+$ and PD-1$^-$ subsets and CD39$^+$ and CD39$^-$ subsets, suggesting that within tumors the activation status of the CD8$^+$ T cells is not a major predictor of $A_{2A}R$ expression. Therefore, whilst TCR activation appears to be a major driver of $A_{2A}R$ expression in CD8$^+$ T cells within draining lymph nodes other factors appear to govern the expression of $A_{2A}R$ within tumors. Characterization of CD8$^+$ T cells by differentiation status revealed that $A_{2A}R$ expression was modestly but significantly higher within the CD69$^+$SLAMF6$^+$ precursor exhausted subset. This is intriguing given previous studies have indicated that $A_{2A}R$ signaling is required for the maintenance of naïve cells, partly due to its ability to upregulate IL-7R expression[16,29,46,47]. Whether $A_{2A}R$ signaling is required for the maintenance and/or expansion of CD69$^+$SLAMF6$^+$ precursor exhausted CD8$^+$ T cells within tumors remains to be determined but is one potential mechanism by which $A_{2A}R$ signaling may actually promote a favorable differentiation status for responses to immune checkpoint blockade in some contexts. It will be interesting in follow-up studies to investigate factors that may influence $A_{2A}R$ expression such as hypoxia and other local immunosuppressive metabolites found at high concentrations within tumors[48]. For example, systemic oxygenation has been shown to decrease expression of CD73, $A_{2A}R$ and $A_{2B}R$ and improve anti-tumor immunity although the cell types that reduced $A_{2A}R$ expression under these conditions could not be determined in these studies as the determination was based upon

mRNA analysis of bulk tumor tissue[6,49]. Amongst CD4$^+$ cells, $A_{2A}R$ expression was highly abundant on CD39$^+$PD-1$^+$ cells, potentially indicative of Treg cells that are known to express these immune checkpoints at high levels within tumors. However, this could not be definitively confirmed due to an inability to counterstain with a Foxp3-directed antibody due to loss of GFP signal upon nuclear permeabilization. Future studies may confirm this by FACS sorting GFP$^+$ and GFP$^-$ cells and then performing analysis of transcription factor expression in each population.

One interesting observation from these analyses is that a subset of cDC2 cells was identified that expressed $A_{2A}R$. These cells exhibited higher expression of MHCII, CD80 and CD86 relative to GFP$^-$ counterparts, indicating that they may represent an activated subset of cDC2. Indeed, consistent with this, we observed that activation of cDC2s in vitro with poly IC led to a significant increase in $A_{2A}R$ expression and in vivo, treatment with immune checkpoint blockade, particularly anti-CTLA-4, led to a significant increase in $A_{2A}R$ expression on cDC2s. Expression of $A_{2A}R$ and $A_{2B}R$ on dendritic cells has been reported previously[50] and blockade of $A_{2B}R$ or CD73[9,51] or myeloid-specific deletion of $A_{2A}R$ or $A_{2B}R$ has been shown to enhance anti-tumor immunity[27,52] in part through modulation of DC function[53]. More recently, it was shown that a component of the mechanism of action for the small molecule $A_{2A}R$ antagonist AZD4635 was enhancement of cDC1 function[54]. To our knowledge the expression of $A_{2A}R$ on cDC2 cells has not been previously reported. Interestingly we observed that $A_{2A}R$ expression was largely restricted to a subset of cDC2 with increased expression of MHCII, CD80 and CD86 that was increased upon treatment with anti-CTLA-4. $A_{2A}R$ may therefore represent a negative feedback loop that limits the costimulatory capacity of activated cDC2s. cDC2s have been shown to be critical in priming intra-tumoral CD4$^+$ T cell responses and so it is interesting to postulate that $A_{2A}R$ expression on these cells may consequently limit the formation of these protective responses[39]. In this study by Binnewies et al. it was shown that Tregs limit the activation and differentiation of cDC2s, which may be linked to the activation and increased expression of $A_{2A}R$ observed in our study following anti-CTLA-4 blockade given we used the 9H10 clone that is known to induce Treg depletion. Therefore, in future studies, it would be of interest to isolate $A_{2A}R^+$ and $A_{2A}R^-$ cDC2s and compare their functional capacity to prime CD8$^+$ and CD4$^+$ T cell responses ex vivo.

Although we observed increases in $A_{2A}R$ expression in cDC2s and, to a lesser extent, cDC1s following immune checkpoint blockade, it was somewhat surprising that we did not observe increases in $A_{2A}R$ expression within tumor-infiltrating T cells following PD-L1 blockade or treatment with anti-CTLA-4 and anti-PD-L1. Previously, an assessment of $A_{2A}R$ mRNA following PD-1 blockade revealed an approximately 2-3 fold increase in $A_{2A}R$ expression within tumor-infiltrating CD8$^+$ T cells[8]. However, we did not observe this in our current study

using the $A_{2A}R$ eGFP reporter mice. The reasons for these differences are not clear but may be related to differences in kinetics in the two studies, the use of anti-PD-L1 as opposed to anti-PD-1 or a true difference between mRNA and protein. Nevertheless, we did observe an increase in $A_{2A}R$ expression within $CD8^+$ T cells isolated from the draining lymph nodes of mice undergoing immune checkpoint blockade, highlighting the potential for $A_{2A}R$ blockade to enhance T cell priming in tertiary lymphoid structures beyond the tumor microenvironment in the context of immunotherapy. Moreover, it is likely that both immune checkpoint blockade and chemotherapy result in increased levels of extracellular adenosine due to the destruction of tumor cells, which in itself may make targeting the adenosine pathway therapeutically more relevant in this context. Interestingly, in the context of immune checkpoint blockade we observed that therapeutic efficacy of anti-PD-L1 was associated with the emergence of a $PD-1^+$ $A_{2A}R^-$ subset that may represent a biomarker for response.

Assessment of the transcriptional profile of $A_{2A}R^+$ and $A_{2A}R^-$ $CD8^+$ T cells revealed that $A_{2A}R^+$ cells elicited a reduced expression of a defined set of STAT5 target genes[40]. This is consistent with previous observations we have made with $CD8^+$ T cells treated in vitro with the adenosine mimetic NECA[16]. This may be linked to the observation that IL-2 and IL-7, cytokines that signal through STAT5, can overcome adenosine-mediated suppression[41,55]. Interestingly, we observed that PD-L1 blockade resulted in increased expression of STAT5 target genes within both $A_{2A}R^+$ and $A_{2A}R^-$ subsets, suggesting that this may help $CD8^+$ T cells to overcome adenosine-mediated suppression through upregulation of this pathway.

Lastly, to utilize this model to identify rational combination therapies we interrogated the impact of various cytokines on $A_{2A}R$ expression as there is remarkably little known about the control of $A_{2A}R$ expression in this regard. Our analysis identified the IL1 family of cytokines, and particularly IL-18, as a potent inducer of $A_{2A}R$ expression on $CD8^+$ T cells, NK cells, $\gamma\delta$T cells and NKT cells. Given that IL-18 is known to induce anti-tumor immunity, but based upon these data may also concomitantly increase $A_{2A}R$ expression, we hypothesized that suppression mediated by $A_{2A}R$ signaling may limit the overall therapeutic effect of IL-18. We therefore evaluated the potential of combining IL-18 therapy with $A_{2A}R$ blockade through engineering tumor cells to secrete IL-18. These experiments confirmed that IL-18 was a strong inducer of $A_{2A}R$ expression in vivo, and anti-tumor effects were even greater in $A_{2A}R$ knockout mice, highlighting the potential for combination therapy targeting the $A_{2A}R$ in conjunction with IL-18 therapy. There has been significant work in developing a form of IL-18 suitable for clinical application, including the development of a form of IL-18 that does not bind to IL-18DR and therefore elicits greater anti-tumor potency than wild-type IL-18[42]. It would therefore be of significant interest to interrogate a possible combination strategy utilizing these such reagents and $A_{2A}R$ antagonists that are currently under clinical evaluation.

## Methods
This research and all study protocols have been approved and comply with the Peter MacCallum Animal Experimental Ethics Committee (AEEC) ethical regulations regarding the use of animals. Studies utilizing human PBMCs from healthy donors was approved by the Peter MacCallum Cancer Centre Human Research Ethics committee. Informed consent was obtained from the Australian Red Cross.

### Animal models and tumor models
$A_{2A}R$ eGFP reporter mice (Tg(Adora2a-EGFP)EP141Gsat) were obtained from The Jackson Laboratory repository and backcrossed onto a C57BL/6 background. The transgene for these mice is the cDNA of EGFP, followed by a polyadenylation signal, inserted into the mouse genomic bacterial artificial chromosome (BAC) RP24-238K3. This insertion is at the start codon of the Adora2a gene so that expression of the reporter mRNA is controlled by the regulatory sequences of *Adora2a*. C57BL/6 wildtype mice were purchased from Australian Bioresources (Moss Vale, New South Wales) or bred in house at the Peter MacCallum Cancer Centre. C57BL/6 $A_{2A}R$ eGFP reporter mice or $A_{2A}R^{-/-}$ mice were bred in house at the Peter MacCallum Cancer Centre. Mice used in experiments were between 6 to 16 weeks of age and experiments were approved by the Animal Experimentation Ethics Committee #E672. AT3 tumor cells were obtained from Dr. Trina Stewart and engineered to express chicken ovalbumin as described previously[8]. MC38 tumor cells were obtained from Dr. Nicole Haynes. All studies using the breast cancer cell lines AT3ova or E0771 were performed in female mice. Experiments with MC38 tumors were performed in male mice unless specified in the source data file. AT3-Her2, E0771-Her2 and MC38-Her2 were generated as previously described[16]. Tumor lines were verified to be *Mycoplasma* negative by PCR analysis and were actively passaged for less than 6 months. Tumor cells were grown in DMEM supplemented with 10% FCS, Glutamax, and penicillin/ streptomycin. For in vivo experiments, the indicated number of cells were resuspended in PBS and injected subcutaneously (100 μL). Tumors were measured using callipers and calculated as mm2 (width*length). Tumor size did not exceed the maximal size approved by the AEEC ethics committee of 150 mm$^2$. To engineer AT3ova and MC38 cells to express IL-18, IL-18 cDNA (ATGAACTTTGGCCGACTTC ACTGTACAACCGCAGTAATACGGAATATAAATGACCAAGTTCTCTTCG TTGACAAAAGACAGCCTGTGTTCGAGGATATGACTGATATTGATCAAA GTGCCAGTGAACCCCAGACCAGACTGATAATATACATGTACAAAGAC AGTGAAGTAAGAGGACTGGCTGTGACCCTCTCTGTGAAGGATAGTAA AATGTCTACCCTCTCCTGTAAGAACAAGATCATTTCCTTTGAGGAAAT GGATCCACCTGAAAATATTGATGATATACAAAGTGATCTCATATTCTT TCAGAAACGTGTTCCAGGACACAACAAGATGGAGTTTGAATCTTCAC TGTATGAAGGACACTTTCTTGCTTGCCAAAAGGAAGATGATGCTTTCA AACTCATTCTGAAAAAAAAGGATGAAAATGGGGATAAATCTGTAATGT TCACTCTCACTAACTTACATCAAAGTTAG) was cloned into the MSCV Cherry vector and retrovirus generated from HEK293gp cells prior to transduction of target cells. mCherry$^+$ cells were FACS sorted prior to functional experiments.

### Antibodies, pharmacological agents, and cytokines
Isotype control (2A3, catalogue number BE0089), anti-PD-L1 (clone 10 F.9G2, catalogue number BE0101) and anti-CTLA-4 (9H10, catalogue number BE0131) were purchased from BioXcell. Mice were treated every 4 days for up to 2 doses. Where indicated mice were treated at days 14, 16, 18 and 20 post-tumor inoculation with 25 μg FTY720 (Sigma). Anti-asialo GM-1 (catalogue number 986-10001, FUJIFILM Wako Pure Chemical Corporation) and anti-CD8α (clone YTS 169.4, catalogue number BE0117, BioXcell) were used for in vivo NK cell and $CD8^+$ T cell depletion, respectively. Anti-CD3 (clone 145-2C11) and anti-CD28 (clone 37.51) used to stimulate murine T cells were obtained from BD Pharmingen. Cytokines used for the stimulation of murine or Human immune cells were purchased from Biolegend.

### Immunofluorescence analysis
$A_{2A}R$ eGFP mice were subcutaneously engrafted with $5 \times 10^5$ AT3-ova tumor cells. Mice were treated with anti-PD-L1 (clone B7-H1) and anti-CTLA-4 (clone 9H10) on days 14 and 18 post-tumor inoculation. Tumors and spleens harvested on Day 20 were fixed with 4% paraformaldehyde (PFA) for 3 hours at 4°C, followed by an overnight incubation in a 30% sucrose PBS solution. Tissues were then embedded in O.C.T compound (Scigen) in cryomolds, stored at -80°C, and serially sectioned at 12μm per tissue slide.

Slides were dried at room temperature and fixed for 5 min in ice-cold acetone. Tissues were blocked with 0.2% bovine serum albumin (BSA) for 5 min and incubated overnight at 4°C with fluorophore-conjugated antibodies prepared at 1:200 dilution in BSA: αCD45.2 AF647 Biolegend, clone: 104, CAT:109818) and αGFP AF488

(Invitrogen, REF: A21311)]. The following day, slides were washed twice in PBS for 5 minutes each at room temperature and were coverslipped with ProLong Gold Antifade mounting medium.

Images were acquired with an Olympus DP80 camera on an Olympus BX53 microscope using the cellSens Dimension program. Images were analyzed with ImageJ.

## Flow cytometry analysis of immune cells ex vivo
The tumors and draining lymph nodes of mice were isolated at the indicated timepoint post-tumor inoculation. Tumors were digested with collagenase type IV (Sigma-Aldrich) and 0.02 mg/ml DNAase (Sigma-Aldrich) for 30 minutes at 37 °C. Cells were then passed through a 70 μm filter twice. Cells were then incubated in Fc Block (supernatant from 2.4G2 hybridoma) and then stained with indicated flow cytometry antibodies and Fixable Yellow (Thermo Fischer Scientific) as a viability dye. αGal Cer loaded tetramer used for the identification of NKT cells was obtained from Dr. Hui-Fern Koay and Prof. Dale Godfrey. Flow cytometry antibodies used in the study are listed in Supplementary Table 1.

## Human NK cell purification from PBMCs
Peripheral blood mononuclear cells (PBMCs) were obtained by performing a Ficoll separation on human donor blood (Source: Australian Red Cross LifeBlood). Red blood cells were lysed with ACK lysis buffer (8.29 g NH2Cl, 1 g KHC3, 0.04 g Na2 EDTA in 1 L sterile MqH₂0 filtered through a 22 μM Stericup vacuum filter (Merck)). NK cells were subsequently washed with PBS and purified using an NK isolation kit (Human NK Cell Isolation Kit, LS Columns, Miltenyi Biotec) as per the manufacturer's instructions. Purified NK cells were washed with PBS and resuspended in culture media (RPMI media + 10% FBS, Sodium pyruvate, NEAA, Glutamax, HEPES, Penicillin/Streptomycin) prior to assay setup.

## Cytokine stimulation assay
Cytokine stimulation assays were performed in 96 well plates. $1 \times 10^5$ NKs were plated in RPMI media (as above). Human IL-18 (50 ng/ml) (R&D systems) and 20 IU/ml IL-2 (NIH) were added to NK cells and plates incubated at 37 °C. NK cells were then spun down at 1400 rpm for 4 min and cell pellets frozen in RLT buffer (Qiagen RNEasy kit) + 10 μl/ml 2-Mercaptoethanol (BME) and stored at −80 °C.

## Quantitative real-time PCR analysis
RNA was isolated using an RNeasy Mini kit (Qiagen) following which cDNA was generated using m-MLV reverse transcriptase (Promega). qRT-PCR was then performed using murine or Human $A_{2A}R$ mRNA probes (Taqman; Hs00169123_m1) as per the manufacturer's instructions with GAPDH (Taqman; Hs02786624_g1) used as a housekeeping gene. For murine cells, L32 was used as a housekeeping gene as per previous work[8].

## Generation of anti-Her2 CAR T cells
Murine splenocytes were activated with anti-CD3/ anti-CD28 before transduction with supernatants derived from a GP + E86 anti-Her2 CAR packaging line as previously described[16]. Briefly, supernatants were added to retronectin (10 μg/ml Takara Bio) coated 6 well plates and spun at 1200 g for 30 minutes. Subsequently 1 ml T cells were added to each well to give a final volume of 10e6 T cells per well. The cells were then spun at the same speed for 90 minutes. This process was repeated 24 h later and after transduction, CAR T cells were maintained in IL-2 and IL-7 prior to coculture with tumor cells at day 6–7 post-activation.

## Gene expression analysis
Using the Quant-seq 3' mRNA-seq Library Prep Kit from Illumina (Lexogen), RNA-seq libraries were prepared from RNA extracted from sorted cells. 75 bp Single-end RNA-sequencing was then performed using NextSeq (Illumina, Inc., San Diego, CA) and base calling was performed using CASAVA 1.8.2. Random primer bias removal and 3' poly-A-tail trimming was performed using Cutadapt v2.1 to derived raw sequences. Quality control was assessed using FastQC v0.11.6 and RNA-SeQC v1.1.8[56]. Next, sequence alignment against the mouse reference genome mm10 was performed using HISAT2. FeatureCounts (Rsubread v2.10.5) was used to count raw reads and genes were then annotated from the Ensembl releases[57]. Normalization of gene counts was performed using the using the EdgeR package[58,59] and the TMM (trimmed means of M-values) method into log2 counts per million (CPM). The quasi-likelihood F test statistical test method based on the generalized linear model (glm) framework was used for differential gene expression (DEGs) comparisons. Adjusted p values were computed using the Benjamini-Hochberg method. Principal component analysis (PCA) was performed generated based on the topmost variable genes. DEGs were classified as significant based on a false discovery rate (FDR) cutoff of less than 0.05. For heatmaps, the pheatmap R package was used to plot row mean centered and scaled normalized log2(CPM + 0.5) values. Genes, columns or rows were sorted by hierarchical clustering using Euclidean distance and average-linkage.

Unbiased gene set enrichment analysis (GSEA) using the fgsea package was performed on fold change ranked DEGs with 1000 permutations (nominal *P*-value cutoff <0.05). Reference gene sets were obtained from the Hallmarks dataset from the MsigDB library or previously published analyses of a STAT5 signature (GEO:GSE41819).

## Statistical analysis
Statistical analyses were performed with one-way ANOVA, two-way ANOVA or unpaired *t* test where appropriate as indicated in the figure legend. $P < 0.05$ was considered significant. For survival curve analysis, mice that developed ulcerated tumors prior to the 100 mm² point were censored from the analysis

## Reporting summary
Further information on research design is available in the Nature Portfolio Reporting Summary linked to this article.

## Data availability
The RNA-sequencing data used in the study have been deposited in the Gene Expression Omnibus under the accession code GSE230135 available at. The remaining data are available within the Article, Supplementary Information or Source Data file. Source data are provided with this paper.

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

## Acknowledgements

The authors would like to acknowledge Prof. Dale Godfrey who provide access to the αGal-Cer tetramer used in the study. This work was funded by a Program Grant from the National Health and Medical Research Council (NHMRC; Grant number 1132373). P.A.B. was supported by a National Breast Cancer Foundation Fellowship (ID# ECF-17-005, 2017-2020) and a Victorian Cancer Agency Mid-Career Fellowship (2021-Current). I.G. House was supported by a Victorian Cancer Agency Early Career Fellowship (ECRF20017). P.K.D. was supported by an NHMRC Senior Research Fellowship (APP1136680). Junyun Lai was a recipient of a US Cancer Research Institute Irvington postdoctoral fellowship (award no. 3530). The authors wish to acknowledge the contributions of Ms Karen Gill, Mr Mike Rear and Mr Graeme Sissing who act as consumer representatives for the laboratory.

## Author contributions

K.L.T. designed and performed the majority of experiments, analyzed the data and wrote the manuscript. J.Lai, Y.K.H., D.M.N., E.B.D., D.N., and C.W.C. designed and performed experiments. K.S. analyzed data and generated scripts for RNA-sequencing analysis. H.F.K. provided reagents. K.S., T.X.H., E.V.P., I.M., I.G.H., J.N.L., J.S.K., J.Li, J.T., M.N.D.M., C.M.S., K.M.Y., A.X.Y.C., P.A.D. and B.H. assisted in the generation of data. I.A.P., R.W.J., and P.K.D. supervised aspects of the study. P.A.B. conceptualized and led the study, designed and performed experiments, analyzed the data and wrote the manuscript.

## Competing interests

P.A.B. declares the following conflicts: research funding from AstraZeneca, Bioardis, Bristol-Myers-Squibb and Gilead Sciences. P.K.D. declares the following conflicts: research funding from Myeloid Therapeutics, Prescient Therapeutics, Bristol-Myers-Squibb and Juno Therapeutics. I.A.P. declares the following conflicts: research funding from Bristol-Myers-Squibb and Astrazeneca. The Johnstone Lab receives research support from Roche, BMS, AstraZeneca and MecRx. R.W.J. is a scientific consultant and shareholder in MecRx. The remaining authors declare no competing interests.
