## [Peer Review File · Nature Communications]

A2AR-eGFP reporter mouse enables elucidation of A2AR expression dynamics during anti-tumor immune responsesREVIEWER COMMENTS

Reviewer #1 (Remarks to the Author): with expertise in cancer immunology, adenosine-mediated immunosuppression

This manuscript by Todd et al. describes an important advance in a fascinating area of research that has direct clinical implications to recent novel approaches to improve cancer immunotherapies. While the fundamental mechanisms of the immunosuppressive hypoxia-adenosinergic pathway have been studied for several decades, promising results from the first clinical studies using inhibitors of this pathway are only recently being reported. Therefore, this work may directly inform current clinical immunotherapeutic strategies. This work comes from leaders in the field and experiments are very well done and clearly described. I believe the manuscript will be of significant interest to the readers of Nature Communications.

The field of hypoxia-adenosinergic immunoregulation has been plagued by the poor performance of antibodies to measure the expression of A2AR. The system developed here is an excellent tool to be able to study this clinically relevant pathway in much more detail. The findings described within the manuscript not only confirm previous mechanism-based observations with this new powerful model, but also uncover very interesting new developments, including the emergence of a PD-1+A2AR⁻ cell population that may correlate with ICB responses and potentially serve as biomarker for therapy.

Overall, this paper is highly interesting, very well written, with only relatively minor concerns (below).

1. Line 64 states that the adenosine that modulates anti-tumor immunity is predominantly produced by CD73/CD39 tandem of ecto-ATPases. Since there are many sources of extracellular adenosine from both outside and inside the cell, this is a bit over-stated and the word predominantly should be reconsidered.

2. References for line 72 and line 74 should include the first reported evidence of anti-tumor effects in A2AR-deficient mice and small molecule antagonists (Ohta et al., PNAS 2006 ◊ Ref

#3)

3. Much of the work presented here focuses on T cells. Was there any characterization done to confirm the transgenic A2AR-GFP T cells retained similar phenotypes and effector functions to A2AR WT control T cells?

4. The following references should be added to line 79 stating that A2AR antagonists enhance efficacy of ACT:

PMCID: PMC1559765 DOI: 10.1073/pnas.0605251103

PMCID: PMC6052792, DOI: 10.4049/jimmunol.1700850

5. Line 121 might also consider referencing earlier studies of A2AR on NK cells (2005) PMID: 16177079 DOI: 10.4049/jimmunol.175.7.4383

6. For Fig. 1F, what is the control group here? What is the baseline expression of A2AR on CART before co-culturing with tumor cells? Since generation of CART requires T cell activation, is A2AR already increased before exposure to tumor cells? This is an important consideration for all CART therapies, since this means that the process required to manufacture CART simultaneously make them highly susceptible to adenosine-mediated suppression. This may help explain why CART cells are ineffective in solid tumors with high levels of extracellular adenosine.

7. Are the important data from Figure 1 and 2 confirmed in a less artificial (non-ova expressing) tumor model?

8. Data from Figure 1B is very interesting. While total MFI appear unchanged in the graph to the right, the flow histograms actually look like two separate populations (A2AR high and low) for the Tetramer - , and one population (A2AR mid/high) for Tetramer +. Was this true in other repetitions of this experiment?

9. Were the authors able to determine the relative expression (e.g. MFI) of A2AR on NK

versus alpha/beta T cells from the tumor? Figure 2F shows the normalized MFI, but is it possible to directly compare NK versus T cells in their expression of A2AR?

10. The data from figure 2H is very interesting and perhaps somewhat counter intuitive. What is the proposed explanation for decreased expression of A2AR on CD8 with advancing tumor progression? Is it possible that more T cells are arriving at these later/larger tumor stages from the DLN (where A2AR expression is lower), and that these early emigrants still have lower A2AR expression compared to the T cells that have been in the TME longer? Or perhaps some of these cells might be reaching an exhausted or apoptotic stage in the TME (less transcriptionally active?). Did authors investigate whether there are any phenotypic differences in these A2AR expressing cells at the early and late time points? Again, it would be useful to confirm this finding in a non ova-expressing tumor cell line.

11. As a general comment, it may be relevant to discuss that while ICB and chemotherapy in this study did not increase expression of A2AR, these therapies likely do increase levels of extracellular adenosine due to destruction of tumor/stromal cells. A2BR may also find a role here when adenosine levels become significantly increased.

12. Line 372 suggests TME hypoxia may alter A2AR of expression. Authors may consider discussing here (or referencing) studies that have documented that the reversal of tumor hypoxia by oxygenation reduces the expression of A2AR/A2BR, CD73, and CD39 as well levels of extracellular adenosine.

PMCID: PMC4641038 DOI: 10.1126/scitranslmed.aaa1260

PMCID: PMC4247798 DOI: 10.1007/s00109-014-1189-3

13. In Figure 4, GFP signal is very low. 20x images would be extremely helpful here. Are the data from B-E on PD1/A2AR- T cells confirmed by flow cytometry?

14. In Figure 6G there are no units on the Y axis. The effect of A2AR appears very minimal in this model. It is possible that adenosine levels are lower in the MC38 model. Additionally,

the treatment starts very early, perhaps when hypoxia/adenosine levels are lower (even though data in the ova-expressing model indicate high A2AR expression on T cells at this stage). Was the experiment ended with the tumor sizes still so small? Was another study done looking at extended growth kinetics (e.g. growth kinetics from the survival experiment in Figure 6H)?

Reviewer #2 (Remarks to the Author): with expertise in cancer immunology, adenosine-mediated immunosuppression

The manuscript presented by Todd and collaborators characterizes the expression of adenosine A2A receptor (A2AR) on different populations of tumor infiltrating lymphocytes and presents evidence supporting the rationale for the use of combined immunotherapies that target A2AR. For this, they develop a novel reporter A2AR mice in C57BL/6 background. The manuscript presents interesting evidence that has been elusive due to the lack of good antibodies against A2AR. In opinion of this reviewer, the data presented is significant to the field but the main weakness of the manuscript is regarding the interpretation of data presented in figure 3. The issues that should be addressed before considering publication are:

Major:

Figures 3A, B, C and 6F – the authors present normalized MFIs, however it is not clear how the normalization was calculated. Related also to the normalization method, the data from the histogram presented in Figure 3C does not seem to be representative as the CD8+SLAMF6-CD69+ population seems to present higher A2AR MFI than the CD8+SLAMF6+CD69+ population. Moreover, in this type of distribution, it is not clear why the authors decide to analyze GFP MFI rather than the frequency of GFP negative and GFP positive populations. Staining of WT control should be included also in Fig 3C. In figure 3B, the tetramer+ population has a unimodal distribution for GFP, whereas the tetramer negative populations seem to have a bimodal distribution, with GFP negative and positive populations. Again, this is not represented in the normalized graph. Thus, the statement “no relationship was observed between tumor-antigen specificity and A2AR expression within

tumor-infiltrating CD8 +T cells” does not seem to be clearly supported by the data presented.

Lines 205-208 - Please provide a stronger rationale for the use of FTY720 and to why the authors expect an increase in A2AR MFI in the presence of treatment with FTY720.

In line 210 the authors indicate that “Within tumors, FTY720 treatment resulted in a reduction in the proportion of CD69+ SLAMF6 + (less-differentiated population) CD8 + T cells and A2AR expression was significantly reduced (Figure 3D) however the evidence of the reduction in the proportion of CD69+SLAMF6 +(less-differentiated population) CD8 +T cells with FTY720 is not presented.

In lines 240-242 the authors state that “whilst the overall frequency of CD8+A2AR +(but not CD4+ A2AR+) T cells was increased by therapy (Figure 4B), the expression of A2AR on a per cell basis was not significantly modulated by these treatment regimens at day 7 post therapy (Figure 4C)”. Is this observation explained by an increase in the frequency of total CD8+ T cells or due to an increase in the percentage of the CD69+ SLAMF6+ (less-differentiated population) CD8 + T cells within tumors after therapy? The authors should clarify this issue.

Minor:

Line 76 – It should say “Renal cell carcinoma” not Renal canal carcinoma. Also, please add a comma after “breast”

Line 131 – The authors should offer an explanation as to why different tumor cell lines induce different levels of A2AR expression in CAR T cells.

Line 142 – gating strategy is presented in Supplementary Figure 1F, not 1A. Supplementary figure 1 should be reorganized as it is cited.

Line 151 – NK cells are considered a % of T cells

Figure 2G (raw data from FACS experiments) should be presented before 2D

As the FLT3L BMDC cultures are very heterogeneous, cDC2 should be characterized based on CD11c/CD11b/MHC-II/Sirpa/XCR1 expression. Are cDC2 in figure 2J identified based on Sirpa also? Are the populations analyzed in figure 2J and supplementary Figure 3 obtained from tumors? As most of the results are related to cDC2, which are not necessarily the main population of DCs involved in cross presentation, how is this result related to the rest of the data regarding CD8+ T cells?

In line 182 the authors indicate “This analysis included expression of markers associated with differentiation status (CD62L, SLAMF6, CD69, CD44), the immune checkpoint PD-1 and CD39 and CD73, the ectoenzymes responsible for the breakdown of ATP to adenosine” however in Figure 3, there is no reference to CD73.

FACS plots from Figure 3A and 3B present different X axis (CD44 or CD62L), however in line 188 the authors indicate “We first investigated the expression of A2AR on CD62L+, CD62L-CD44 SIINFEKL tetramer +and CD62L-CD44 +SIINFEKL tetramer -cells”.

In line 244 when the authors indicate “To further interrogate this” do they mean carboplatin therapy?

Please define therapeutic effect in Figure 4E and responder/non responder in Figure 4F

Figure 5A is difficult to interpret due to the color palette selected. As pink-purple and orange-green dots are part of a same treatment, it is not clear why there are different number of replicates (2 and 3) in each group.

Line 290 – the authors indicate that “The association between A2AR+ status and reduced STAT5 target genes was of interest because of our previous observations that A2AR agonists negatively regulate JAK-STAT signaling [16] and a previous report from Cekic and colleagues that IL7R signaling protects CD8 + T cells from adenosine mediated-suppression [39]”, however these observations seem to be in contradiction. Please elaborate further.

Figure 6C- please change histogram colors to identify better different treatments.

In line 365 the authors state: “Moreover, within tumors A2AR expression was similar between PD-1 + and PD-1 - subsets, CD39 + and CD39 - subsets and SLAMF6 + and SLAMF6 - subsets, suggesting that within tumors the activation status of the CD8 +T cells is not a predictor of A2AR expression”, however in line 200 they indicate that “Analysis of these distinct subsets indicated that A2AR was modestly but significantly enriched within the CD69+ SLAMF6 + subset relative to SLAMF6- counterparts (Figure 3C)”. Please explain this contradiction.

Line 488 mentions the use of an antibody against CD4 for immunofluorescence however there is no CD4 staining (immunofluorescence) presented in the manuscript

The material and methods section lacks information regarding FTY720 treatment.

Line 613 – says “that are CD44+Tetramer +or Tetramer”, please add negative (-) to Tetramer

Reviewer #3 (Remarks to the Author): with expertise in cancer immunology, adenosine-mediated immunosuppression

In this study Todd et al. evaluated A2AR expression in different immune cell subsets using a mouse model expressing GFP under A2AR promoter. Using this model as a tool they have also identified dynamics of A2AR expression after establishing some commonly used syngeneic tumor models and after treating the mice with different therapeutic agents for cancer. The study also adds to previous literature that there may be an A2AR/cytokine cross talk rendering tumors more or less responsive to different therapeutic strategies including A2AR blockade. Overall, the model and results are novel and will provide important information to the field. Here are some detailed comments, suggestions, and discussion points:

- Abstract: it is not unexpected that myeloid DCs highly express A2AR. The results seem complementary to the vast literature suggesting APCs express A2AR and they are modulated

by specific A2AR agonists/antagonists...

- Further analysis of CD44 dim/naïve vs high (effector memory) at steady state can be important. Those 20% A2AR+ cells may be memory-like (CD44high) T cells.

- In this study A2AR expression in T cells, especially CD8 T cells shows similar profile as TCF-7 expression, which may highlight some beneficial effects of A2AR signaling in T cells for anti-tumor immunity by keeping them less differentiated/exhausted and metabolically fit for the tumor microenvironment [Mager et al. Science 2020 (DOI: 10.1126/science.abc3421), Roseblatt et al. Frontiers Cell and Dev. Biol. 2021

(<https://doi.org/10.3389/fcell.2021.647058>), Cekic and Linden, Cancer Res. 2014

(<https://doi.org/10.1158/0008-5472.CAN-13-3581>) , Cekic et al. JEM 2013

(<https://doi.org/10.1084/jem.20130249>)]

- On our hands, FoxP3+ CD4 T cells in the tumors are mostly CD39+ or vice versa. And they are co-stained with CD25. A separate staining to confirm this in a naïve mice may convince readers and other reviewers that despite the lack of FoxP3 staining your findings are in fact shows A2AR is very highly expressed in T regs.

- Authors states that since A2AR expression in CD8 T cells peaks early and starts getting reduced, treating patients with A2AR antagonists early will cause more benefit. This needs to be supported by data or further speculations/discussions needed such as early A2AR expression may be important for their maintenance, reduction of A2AR later during the growth may mean TME getting less hypoxic (VGEF, vascularization) etc...

- Making tumors express the cytokine may not be same as treating them. There are reports suggesting MyD88 signaling downstream of IL-1 or IL-18 can be pro-tumoral or anti-tumor depending on the context. IL-18/A2AR hypothesis may better be tested by treating animals with specific antagonists or agonists. Alternatively, showing whether combinatorial effect of A2AR/IL-18 is mainly driven by innate or adaptive immunity (by using scid or rag-/- animals) may add value to this study. (Especially given that most of NK cells and activated cDC2s express A2AR)

- Discussing the results of this study not only for the pro-tumoral properties of A2AR in general but also anti-tumoral properties especially through increasing metabolic adaptation of T cells in the TME can be fairer. Most antagonists in the trials failed or showed very limited success in mono or combo setting. I think the dynamics of T cell expression of A2AR in TME may be telling something.

- The tumor models used in the study are very immunogenic. Testing the dynamics of A2AR expression in tumor models less responsive to IO agents or A2AR blockade (such as B16 melanoma) can add significant value to this study since extremely high or lower than expected expression of A2AR may suggest they are not good targets for these tumors to begin with.
- Please indicate the unit of tumor size in graphs.

Point by point response

We thank each of the reviewers for their considered questions. We believe that the new experimental data that has led to the addition of 20 new Figure panels have significantly improved the strength of the conclusions and the clarity of the manuscript.

Reviewer 1:

Reviewer #1 (Remarks to the Author): with expertise in cancer immunology, adenosine-mediated immunosuppression

This manuscript by Todd et al. describes an important advance in a fascinating area of research that has direct clinical implications to recent novel approaches to improve cancer immunotherapies. While the fundamental mechanisms of the immunosuppressive hypoxia-adenosinergic pathway have been studied for several decades, promising results from the first clinical studies using inhibitors of this pathway are only recently being reported. Therefore, this work may directly inform current clinical immunotherapeutic strategies. This work comes from leaders in the field and experiments are very well done and clearly described. I believe the manuscript will be of significant interest to the readers of Nature Communications.

The field of hypoxia-adenosinergic immunoregulation has been plagued by the poor performance of antibodies to measure the expression of A2AR. The system developed here is an excellent tool to be able to study this clinically relevant pathway in much more detail. The findings described within the manuscript not only confirm previous mechanism-based observations with this new powerful model, but also uncover very interesting new developments, including the emergence of a PD-1+A2AR⁻ cell population that may correlate with ICB responses and potentially serve as biomarker for therapy.

Overall, this paper is highly interesting, very well written, with only relatively minor concerns (below).

Response: We thank the reviewer for their positive evaluation of our work.

1. Line 64 states that the adenosine that modulates anti-tumor immunity is predominantly produced by CD73/CD39 tandem of ecto-ATPases. Since there are many sources of extracellular adenosine from both outside and inside the cell, this is a bit over-stated and the word predominantly should be reconsidered.

Response: We agree with the reviewer and have amended this sentence as follows on **line 64** of the revised manuscript.

“Adenosine is an immunosuppressive metabolite that modulates anti-tumor immunity and is produced from the degradation of adenine nucleotides by the ecto-enzymes CD73 and CD39 on tumor cells, stroma and fibroblasts or through direct export from the intracellular compartment of cells undergoing hypoxia and/or stress [1-7]”.

2. References for line 72 and line 74 should include the first reported evidence of anti-tumor effects in A2AR-deficient mice and small molecule antagonists (Ohta et al., PNAS 2006 □ Ref #3)

Response: We agree with the reviewer and have added this reference at the suggested points.

3. Much of the work presented here focuses on T cells. Was there any characterization done to confirm the transgenic A2AR-GFP T cells retained similar phenotypes and effector functions to A2AR WT control T cells?

Response: To address this we have performed new experiments comparing the capacity of NECA (a pan adenosine agonist) to suppress T cell cytokine production in these two mice strains. This data confirms that both WT and A_{2A}R eGFP splenocytes elicit a similar level of cytokine production following activation and that NECA similarly

suppresses production of TNF following activation of either WT or A_{2A}R eGFP splenocytes. This data is presented in revised **Figure 1G** and referred to in the following text located on **line 132**.

To confirm that A_{2A}R expression in reporter mice was functional we evaluated their response to adenosine receptor stimulation. Suppression of TNF production by NECA, a pan adenosine receptor agonist, and reversal of this phenotype by SCH58261, an A_{2A}R antagonist, was observed following activation of splenocytes from A_{2A}R eGFP reporter mice or wild-type controls (**Figure 1G**).

4. The following references should be added to line 79 stating that A2AR antagonists enhance efficacy of ACT:

PMCID: PMC1559765 DOI: 10.1073/pnas.0605251103

PMCID: PMC6052792, DOI: 10.4049/jimmunol.1700850

Response: We agree with the reviewer and have added these references at the suggested point.

5. Line 121 might also consider referencing earlier studies of A2AR on NK cells (2005) PMID: 16177079 DOI: 10.4049/jimmunol.175.7.4383

Response: We agree with the reviewer and have added this reference at the suggested point.

6. For Fig. 1F, what is the control group here? What is the baseline expression of A2AR on CART before coculturing with tumor cells? Since generation of CART requires T cell activation, is A2AR already increased before exposure to tumor cells? This is an important consideration for all CART therapies, since this means that the process required to manufacture CART simultaneously make them highly susceptible to adenosine-mediated suppression. This may help explain why CART cells are ineffective in solid tumors with high levels of extracellular adenosine.

Response: The control group shown in original **Figure 1F** (Now **Figure 1I** in the revised manuscript) is CAR T cells that were not stimulated with antigen positive tumor cells. That is to say it is the baseline expression of A_{2A}R on CAR T cells before coculturing with tumor cells. The reviewer is correct that the generation of CAR T cells does indeed also increase A_{2A}R expression. Taking into account the reviewer's suggestion we have provided new data (Revised **Figure 1H**) that shows the increased expression of A_{2A}R on CAR T relative to naïve T cells that is further increased following stimulation with antigen expressing tumor cells. This is referred to in the following text on **line 138** of the revised manuscript.

“Generation of CAR T cells led to increased expression of GFP relative to naïve T cells (**Figure 1H**) and coculture of anti-Her2 CAR T cells with AT-3-, E0771-, or MC38-Her2 expressing tumor cells led to a further and significant induction of GFP in both CD8⁺ and CD4⁺ CAR T cells (**Figure 1I**).”

7. Are the important data from Figure 1 and 2 confirmed in a less artificial (non-ova expressing) tumor model?

Response: In the original manuscript we examined A_{2A}R expression in both AT3ova and MC38 models. However, given the comments by reviewer 3 we now provide new data in the E0771 model, which is unresponsive to immune checkpoint blockade. This new data is presented in **Supplementary Figure 2** and discussed at relevant points in the text.

8. Data from Figure 1B is very interesting. While total MFI appear unchanged in the graph to the right, the flow histograms actually look like two separate populations (A2AR high and low) for the Tetramer - , and one population (A2AR mid/high) for Tetramer +. Was this true in other repetitions of this experiment?

Response: We assume the reviewer is referring to Figure 3B and not 1B with this question. We have reanalyzed this data, prompted by comments from reviewer 1 and also reviewer 2. We have performed 4 repeats of this experiment and we consistently observe that the tetramer positive cells have a more uniform expression of A_{2A}R

relative to the tetramer negative subsets. Prompted by Reviewer 2's good suggestion we have now analyzed this data in terms of a percentage of each cell type expressing GFP and this data indicates that the percentage of GFP⁺ cells is significantly increased in the CD62L⁻tetramer⁺ population relative to the CD62L⁻tetramer negative population. This new data is presented in revised **Figure 3B**. We have also included a representative flow plot that depicts GFP expression vs tetramer positivity to illustrate this point in revised **Supplementary Figure 1G**. This new data is referred to in the following text located on **line 202** of the revised manuscript.

“Within tumor draining lymph nodes A_{2A}R expression was significantly upregulated on CD62L⁻tetramer⁺ CD8⁺ T cells relative to other CD8⁺ T cell subsets (**Figure 3A**), consistent with the notion that A_{2A}R was upregulated following activation of these cells by tumor antigens presented by APCs in the draining lymph nodes. However, within the tumors there was only a modest increase in A_{2A}R expression on CD62L⁻ tetramer positive cells relative to CD62L⁻ tetramer negative counterparts (**Figure 3B**; **Supplementary Figure 1G**).”

9. Were the authors able to determine the relative expression (e.g. MFI) of A2AR on NK versus alpha/beta T cells from the tumor? Figure 2F shows the normalized MFI, but is it possible to directly compare NK versus T cells in their expression of A2AR?

Response: To address this we have provided new data in revised **Figure 2H** indicating the A_{2A}R MFI within each subset isolated from AT3ova tumors. This data is referred to in the following text located on **line 166**.

Within the tumors, NK cells expressed a significantly higher level of A_{2A}R than all other immune subsets analyzed on a per cell basis (**Figure 2H**).

10. The data from figure 2H is very interesting and perhaps somewhat counter intuitive. What is the proposed explanation for decreased expression of A2AR on CD8 with advancing tumor progression? Is it possible that more T cells are arriving at these later/larger tumor stages from the DLN (where A2AR expression is lower), and that these early emigrants still have lower A2AR expression compared to the T cells that have been in the TME longer? Or perhaps some of these cells might be reaching an exhausted or apoptotic stage in the TME (less transcriptionally active?). Did authors investigate whether there are any phenotypic differences in these A2AR expressing cells at the early and late time points? Again, it would be useful to confirm this finding in a non ova-expressing tumor cell line.

Response: We thank the reviewer for this interesting question. To address this we investigated further the phenotype of CD8⁺ T cells within this timecourse. One aspect that stood out and at least partly explain the reduced expression of A_{2A}R in CD8⁺ T cells in this tumor model is the reduction in proportion of tetramer positive cells and CD62L⁺ cells during tumor progression. Since both of these cells express more A_{2A}R than CD62L⁻ tetramer negative counterparts, this contributes to the overall reduced expression of A_{2A}R in the total CD8⁺ population. However, this does not fully account for the phenotype since at day 10 both the CD62L⁺ and tetramer⁺ populations express significantly more A_{2A}R than at later timepoints. This new data is shown in **Revised Supplementary Figure 1H** and is referred to in the text below located on **line 218**. Furthermore we have now performed a timecourse in E0771 tumors and observed the same effect. This new data is presented in **Revised Supplementary Figure 2C**.

“We next investigated whether these differences contributed to the decreased expression of A_{2A}R observed in CD8⁺ tumor infiltrating lymphocytes over time (**Figure 2I**). We observed that in the course of tumor progression that the proportion of both the antigen specific tetramer positive population and the SLAMF6⁺CD69⁺ progenitor population were progressively diminished (**Supplementary Figure 1H**). Given that these cell types were the highest expressors of A_{2A}R this partly accounts for the reduced expression of A_{2A}R in total CD8⁺ T cells over time. Moreover, on a per cell basis these cell populations elicited significantly higher expression of A_{2A}R at day 10 than other timepoints but there was no significant difference from day 17 onwards (**Supplementary Figure 1H**). Therefore the reduction in A_{2A}R expression in CD8⁺ T cells over time is partly explained by a reduced frequency of the tetramer positive and SLAMF6⁺CD69⁺ cells and partly due to reduced expression of A_{2A}R by these subsets on a per cell basis at later timepoints.”

11. As a general comment, it may be relevant to discuss that while ICB and chemotherapy in this study did not increase expression of A2AR, these therapies likely do increase levels of extracellular adenosine due to destruction of tumor/stromal cells. A2BR may also find a role here when adenosine levels become significantly increased.

Response: We agree with the reviewer's suggestion, and we have added the following text to **line 462** of the revised manuscript to address this point.

“Moreover, it is likely that both immune checkpoint blockade and chemotherapy result in increased levels of extracellular adenosine due to destruction of tumor cells, which in itself may make targeting the adenosine pathway therapeutically more relevant in this context.”

12. Line 372 suggests TME hypoxia may alter A2AR of expression. Authors may consider discussing here (or referencing) studies that have documented that the reversal of tumor hypoxia by oxygenation reduces the expression of A2AR/A2BR, CD73, and CD39 as well levels of extracellular adenosine.

PMCID: PMC4641038 DOI: 10.1126/scitranslmed.aaa1260

PMCID: PMC4247798 DOI: 10.1007/s00109-014-1189-3

Response: We agree that these points are important, and we have added the suggested references, which are referred to in the extended discussion on this point located on **line 416** of the revised manuscript.

“For example, systemic oxygenation has been shown to decrease expression of CD73, A_{2A}R and A_{2B}R and improve anti-tumor immunity although the cell types that reduced A_{2A}R expression under these conditions could not be determined in these studies as the determination was based upon mRNA analysis of bulk tumor tissue [6,46].”

13. In Figure 4, GFP signal is very low. 20x images would be extremely helpful here. Are the data from B-E on PD1/A2AR- T cells confirmed by flow cytometry?

Response: To address this point we have provided images taken at 20x resolution which is located in revised **Figure 4A**. The data in **Figure 4B-4E** are entirely based upon flow cytometry. We apologize for this confusion and we have amended the text at **line 268** to clarify this point.

14. In Figure 6G there are no units on the Y axis. The effect of A2AR appears very minimal in this model. It is possible that adenosine levels are lower in the MC38 model. Additionally, the treatment starts very early, perhaps when hypoxia/adenosine levels are lower (even though data in the ova-expressing model indicate high A2AR expression on T cells at this stage). Was the experiment ended with the tumor sizes still so small? Was another study done looking at extended growth kinetics (e.g. growth kinetics from the survival experiment in Figure 6H)?

Response: We thank the reviewer for noting the mistake on the y axis of **Figure 6G**. We have now added the unit (mm²) to rectify this issue. We agree that the impact of A_{2A}R deficiency alone is minimal in this model. The reasons for this are not fully understood but it is possible that the levels of adenosine are lower in this model. Interestingly, a comparison of adenosine concentrations and gene signatures across various syngeneic models suggests MC38 is intermediate in this regard (PMID 31953314) although AT3 was not included in this study, The data presented in **Figure 6G** reflects the timepoint at which the first control tumors reach the ethical endpoint of 100mm² but the remaining mice in this experiment were tracked for long-term tumor growth kinetics. These data are presented as the survival analysis in **Figure 6H**. To clarify this point we have provided additional data depicting the long-term tumor growth of each mouse in new **Supplementary Figure 6D**.

Reviewer 2

Reviewer #2 (Remarks to the Author): with expertise in cancer immunology, adenosine-mediated immunosuppression

The manuscript presented by Todd and collaborators characterizes the expression of adenosine A2A receptor (A2AR) on different populations of tumor infiltrating lymphocytes and presents evidence supporting the rationale for the use of combined immunotherapies that target A2AR. For this, they develop a novel reporter A2AR mice in C57BL/6 background. The manuscript presents interesting evidence that has been elusive due to the lack of good antibodies against A2AR. In opinion of this reviewer, the data presented is significant to the field but the main weakness of the manuscript is regarding the interpretation of data presented in figure 3.

Response: We thank the reviewer for their positive evaluation of our work.

The issues that should be addressed before considering publication are:

Major:

1. Figures 3A, B, C and 6F – the authors present normalized MFIs, however it is not clear how the normalization was calculated.

Response: Normalization was performed in order to account for differences in MFI observed between each experiment as a result of changes to the flow cytometer laser settings over time. We apologize that the process of normalization was not clearer. For **Figures 3A-C** the data were normalised to the average MFI of total CD8⁺ T cells within each experiment whilst for **Figure 6F** data were normalized to the average MFI of the relevant cell population in control tumors. To clarify this, additional detail has been included in the figure legends.

2. Related also to the normalization method, the data from the histogram presented in Figure 3C does not seem to be representative as the CD8⁺SLAMF6⁺CD69⁺ population seems to present higher A2AR MFI than the CD8⁺SLAMF6⁺CD69⁻ population.

Response: The data in the histogram is from a concatenated sample based upon each mouse in that repeat of the experiment. We checked the MFI for each population in this sample and the values are as follows SLAMF6⁺CD69⁺ (2108) SLAMF6⁺CD69⁻ (1693), SLAMF6⁻CD69⁻ (1114), SLAMF6⁻CD69⁺ (1676). This is representative of the pooled data presented in **Figure 3C**.

3. Moreover, in this type of distribution, it is not clear why the authors decide to analyze GFP MFI rather than the frequency of GFP negative and GFP positive populations.

Response: We agree that determining the percent of GFP⁺ and GFP⁻ cells is another valid way to analyse this data and we have now presented these data in the respective figure panels in Figure 3 (**Figure 3A-C**)

4. Staining of WT control should be included also in Fig 3C.

Response: We thank the reviewer for this suggestion, and this has now been included.

5. In figure 3B, the tetramer⁺ population has a unimodal distribution for GFP, whereas the tetramer negative populations seem to have a bimodal distribution, with GFP negative and positive populations. Again, this is not represented in the normalized graph. Thus, the statement “no relationship was observed between tumor-antigen specificity and A2AR expression within tumor-infiltrating CD8⁺T cells” does not seem to be clearly supported by the data presented.

Response: We thank the reviewer for this suggestion. We have now included this analysis and by this method the expression of A_{2A}R is significantly higher on CD62L⁻tetramer⁺ cells relative to CD62L⁻ counterparts. Although

the difference is relatively modest in terms of magnitude we have modified our conclusions in relation to this data accordingly. This data is referred to in the following text located on **line 206** in the Results section and **line 399** in the Discussion:

“However, within the tumors there was only a modest increase in A_{2A}R expression on CD62L⁻ tetramer positive cells relative to CD62L⁻ tetramer negative counterparts (**Figure 3B**; **Supplementary Figure 1G**)”.

“Whilst antigen (ova)-specific CD8⁺ T cells expressed significantly higher levels of A_{2A}R than non ova-specific CD8⁺ T cells in draining lymph nodes, this difference was more modest within tumors.”

6. Lines 205-208 - Please provide a stronger rationale for the use of FTY720 and to why the authors expect an increase in A2AR MFI in the presence of treatment with FTY720.

Response: Administration of FTY720 was used as a method to determine whether A_{2A}R expression would increase over time as cells spent more time in the tumor microenvironment, as it prevents lymph node egress and therefore attenuates the arrival of “new” lymphocytes to the tumor. We did not necessarily expect A_{2A}R to be increased, but this is the result that would be predicted if A_{2A}R was more highly expressed on more terminally differentiated cells. To clarify this point we have amended the text on **line 232** as follows:

“We reasoned that if A_{2A}R was expressed more highly on terminally differentiated cells that GFP expression would be increased following FTY720 treatment since this treatment would prevent less differentiated cells egressing from the draining lymph node and replenishing the SLAMF6⁺CD69⁺ population.”

7. In line 210 the authors indicate that “Within tumors, FTY720 treatment resulted in a reduction in the proportion of CD69⁺ SLAMF6⁺ (less-differentiated population) CD8⁺ T cells and A_{2A}R expression was significantly reduced (Figure 3D) however the evidence of the reduction in the proportion of CD69⁺SLAMF6⁺(less-differentiated population) CD8 +T cells with FTY720 is not presented.

Response: This data has now been included in revised **Figure 3D**.

8. In lines 240-242 the authors state that “whilst the overall frequency of CD8+A2AR +(but not CD4+ A2AR+) T cells was increased by therapy (Figure 4B), the expression of A2AR on a per cell basis was not significantly modulated by these treatment regimens at day 7 post therapy (Figure 4C)”. Is this observation explained by an increase in the frequency of total CD8⁺ T cells or due to an increase in the percentage of the CD69⁺ SLAMF6⁺ (less-differentiated population) CD8 + T cells within tumors after therapy? The authors should clarify this issue.

Response: This is due to an increase in the frequency of total CD8⁺ T cells and in fact the proportion of CD69⁺SLAMF6⁺ cells is reduced by immune checkpoint blockade, consistent with the paradigm of PD-1 blockade leading to differentiation of this population. This text has been amended to clarify this issue on **line 268**.

“In terms of T lymphocyte phenotype, whilst the overall frequency of CD8⁺A2AR⁺ (but not CD4⁺A2AR⁺) T cells was increased by therapy due to an increase in the proportion of CD8⁺ T cells as determined by flow cytometry analyses.”

Minor:

9. Line 76 – It should say “Renal cell carcinoma” not Renal canal carcinoma. Also, please add a comma after “breast”

Response: We thank the reviewer for noticing this mistake and have made the suggested corrections.

10. Line 131 – The authors should offer an explanation as to why different tumor cell lines induce different levels of A2AR expression in CAR T cells.

Response: The reasons why the tumor cell lines used in Figure 1 upregulate $A_{2A}R$ on CAR T cells to a different extent are not fully understood. It may be related to the level of Her2 expression on each tumor as MC38-Her2 expresses lower levels of Her2 than AT-3 Her2 and E0771-Her2. However, given we do not have a definitive reason for this and it is not a major focus of the manuscript we prefer not to speculate on this within the manuscript text. However if the reviewers and editor believe this is still important, we would be happy to incorporate this.

11. Line 142 – gating strategy is presented in Supplementary Figure 1F, not 1A. Supplementary figure 1 should be reorganized as it is cited.

Response: We apologize for this mistake and have reordered the Figure accordingly.

Line 151 – NK cells are considered a % of T cells

Response: We thank the reviewer for noting this mistake and have removed the “T” from this sentence.

12. Figure 2G (raw data from FACS experiments) should be presented before 2D

Response: The order of this figure has been rearranged as suggested.

13. As the FLT3L BMDC cultures are very heterogeneous, cDC2 should be characterized based on CD11c/CD11b/MHC-II/Sipra/XCR1 expression.

Response: For in vitro cocultures cDC2s were characterized as $CD45R^-MHCII^+SIRPa^+$. Upon reanalysis we observed that the $SIRPa^+$ contained a minor fraction of $CD24^+$ cells (a preferred marker to detect cDC1 in these in vitro cocultures to XCR1) and so we added an additional gate for $CD24^-$ for the re-analysis. This gating strategy is shown below for the reviewer’s interest and can be included as a **Supplementary Figure** if requested. The re-analysed data is now shown in **Supplementary Figure 4C-D**. The data are very similar to what was presented in the original manuscript and the same result was obtained i.e. poly IC stimulation of these cultures leads to a significant induction of $A_{2A}R$ expression on cDC2 cells.

14. Are cDC2 in figure 2J identified based on Sirpa also? Are the populations analyzed in figure 2J and supplementary Figure 3 obtained from tumors?

Response: The populations shown in **Figure 2J** are derived from AT3ova tumors and gated as $CD45^+TCR\beta^-NK1.1^-Ly6C^-CD64^-CD11c^+MHCII^+F4/80lowCD11b^+CD103^-XCR1^-$ as indicated in the figure legend. Data in **Supplementary Figure 3** (now **Supplementary Figure 4**) refer to cDC2 gated from BMDM cultures as noted above.

15. As most of the results are related to cDC2, which are not necessarily the main population of DCs involved in cross presentation, how is this result related to the rest of the data regarding $CD8^+$ T cells?

Response: Although our manuscript has characterized A_{2A}R expression on CD8⁺ T cells we do not doubt the role of other immune populations on anti-tumor immunity. Thus we believe the observation of A_{2A}R expression on activated cDC2s is of significant interest regardless of its impact on CD8⁺ T cell responses. As noted on **line 442** of our manuscript cDC2s have been shown to play an important role in priming CD4⁺ T cell responses, representing just one mechanism by which cDC2 cells can influence anti-tumor immunity. Thus, suppression of these cells by adenosine has the potential to limit anti-tumor immunity independently of direct effects on CD8⁺ T cells.

16. In line 182 the authors indicate “This analysis included expression of markers associated with differentiation status (CD62L, SLAMF6, CD69, CD44), the immune checkpoint PD-1 and CD39 and CD73, the ectoenzymes responsible for the breakdown of ATP to adenosine” however in Figure 3, there is no reference to CD73.

Response: We thank the reviewer for this observation. We have removed “CD73” from this sentence.

17. FACS plots from Figure 3A and 3B present different X axis (CD44 or CD62L), however in line 188 the authors indicate “We first investigated the expression of A2AR on CD62L+, CD62L-CD44 SIINFEKL tetramer +and CD62L-CD44 +SIINFEKL tetramer -cells”.

Response: We thank the reviewer for this comment and have amended the text as suggested.

18. In line 244 when the authors indicate “To further interrogate this” do they mean carboplatin therapy?

Response: We apologize for the confusion, this statement refers to responses following immune checkpoint blockade. To clarify this we have amended this sentence on **line 273** as follows.

“To further interrogate the impact of immune checkpoint blockade on A_{2A}R expression, and to investigate the possibility that A_{2A}R was transiently upregulated following treatment, further experiments were performed to determine the expression of GFP on CD8⁺ TILs at days 2 and days 4 post treatment with either anti-PD-L1 or anti-PD-L1 and anti-CTLA-4.”

19. Please define therapeutic effect in Figure 4E and responder/non responder in Figure 4F.

Response: The percentage therapeutic effect was calculated with the following formula, which has now been added to the figure legend. Therapeutic efficacy = 100-(tumor weight in test sample/average tumor weight in control group*100). The data points selected to represent responders and non-responders are now shown in red in revised **Figure 4E** and this information added to the Figure legend.

20. Figure 5A is difficult to interpret due to the color palette selected. As pink-purple and orange-green dots are part of a same treatment, it is not clear why there are different number of replicates (2 and 3) in each group.

Response: We have amended the color palette as suggested. We submitted triplicate samples for each condition for RNA-Sequencing however there was technical problems with 2 of the samples. These samples were excluded based on the fastQC report, which indicated a high % Adenine indicating possible contamination and a reduced per base sequence quality for those two samples.

21. Line 290 – the authors indicate that “The association between A2AR+ status and reduced STAT5 target genes was of interest because of our previous observations that A2AR agonists negatively regulate JAK-STAT signaling [16] and a previous report from Cekic and colleagues that IL7R signaling protects CD8⁺ T cells from adenosine mediated-suppression [39]”, however these observations seem to be in contradiction. Please elaborate further.

Response: Activation of A_{2A}R has been shown to inhibit JAK/STAT signaling through the induction of SOCS3 (PMID 16914720). IL7 expression was shown by Cekic and colleagues to overcome the effects of adenosine mediated expression. In this study this was attributed to the suppression of FOXO1 activity by IL7R, which countered the increased FOXO1 activity mediated by adenosine, however IL7 signaling also activates STAT5 and

so this is a possible alternative or complementary mechanism that is yet to be explored. To elaborate on this we have amended the text on **line 324** as follows:

“Although Cekic *et al.* attributed the impact of IL7 signaling on adenosine-mediated suppression to the inactivation of FOXO1, the activation of STAT5 signaling by IL7 represents a possible complementary mechanism of action.”

22. Figure 6C- please change histogram colors to identify better different treatments.

Response: This has been amended as suggested.

23. In line 365 the authors state: “Moreover, within tumors A2AR expression was similar between PD-1 + and PD-1 - subsets, CD39 + and CD39 - subsets and SLAMF6 + and SLAMF6 - subsets, suggesting that within tumors the activation status of the CD8 +T cells is not a predictor of A2AR expression”, however in line 200 they indicate that “Analysis of these distinct subsets indicated that A2AR was modestly but significantly enriched within the CD69+ SLAMF6 + subset relative to SLAMF6- counterparts (Figure 3C)”. Please explain this contradiction.

Response: Although there was a statistically significant difference between SLAMF6⁺CD69⁺ cells and SLAMF6⁻ cells, the magnitude of this effect was relatively modest and therefore our interpretation was that the expression of A_{2A}R between the two subsets is “similar”. However, to clarify this point we have removed reference to SLAMF6 at this juncture in the text.

24. Line 488 mentions the use of an antibody against CD4 for immunofluorescence however there is no CD4 staining (immunofluorescence) presented in the manuscript.

Response: We thank the reviewer for this observation and have removed reference to the CD4 directed antibody.

25. The material and methods section lacks information regarding FTY720 treatment.

Response: This information was contained within the figure legend but for clarity this information has also been added to **line 537** of the Materials and Methods.

26. Line 613 – says “that are CD44+Tetramer +or Tetramer”, please add negative (-) to Tetramer

Response: We thank the reviewer for this observation and have amended as suggested.

Reviewer 3

In this study Todd et al. evaluated A2AR expression in different immune cell subsets using a mouse model expressing GFP under A2AR promoter. Using this model as a tool they have also identified dynamics of A2AR expression after establishing some commonly used syngeneic tumor models and after treating the mice with different therapeutic agents for cancer. The study also adds to previous literature that there may be an A2AR/cytokine cross talk rendering tumors more or less responsive to different therapeutic strategies including A2AR blockade. Overall, the model and results are novel and will provide important information to the field.

Response: We thank the reviewer for their positive evaluation of our work.

Here are some detailed comments, suggestions, and discussion points:

1. Abstract: it is not unexpected that myeloid DCs highly express A2AR. The results seem complementary to the vast literature suggesting APCs express A2AR and they are modulated by specific A2AR agonists/antagonists...

Response: We take this comment on board and have revised the abstract accordingly.

2. Further analysis of CD44 dim/naïve vs high (effector memory) at steady state can be important. Those 20% A_{2A}R⁺ cells may be memory-like (CD44^{high}) T cells.

Response: We have performed this analysis and the reviewer is correct in their supposition that the majority of A_{2A}R⁺ cells in the spleen are CD44⁺ cells. This new data is shown in revised **Figure 1C** and referred to in the following text on **line 121** of the revised manuscript.

“Flow cytometry analyses of splenocytes revealed that GFP expression was highest in NK cells that are known to express high levels of A_{2A}R (**Figure 1B**) [24, 28] and CD8⁺CD44⁺ T cells, which is also consistent with a previous report [29] (**Figure 1C**).”

3. In this study A_{2A}R expression in T cells, especially CD8 T cells shows similar profile as TCF-7 expression, which may highlight some beneficial effects of A_{2A}R signaling in T cells for anti-tumor immunity by keeping them less differentiated/exhausted and metabolically fit for the tumor microenvironment [Mager et al. Science 2020 (DOI: 10.1126/science.abc3421), Roseblatt et al. Frontiers Cell and Dev. Biol. 2021 (<https://doi.org/10.3389/fcell.2021.647058>), Cekic and Linden, Cancer Res. 2014 (<https://doi.org/10.1158/0008-5472.CAN-13-3581>), Cekic et al. JEM 2013 (<https://doi.org/10.1084/jem.20130249>)]

Response: We appreciate the suggestion to include discussion of these studies. We have cited this work in an extended discussion in the following text on **line 406** of the revised manuscript.

“Characterization of CD8⁺ T cells by differentiation status revealed that A_{2A}R expression was modestly but significantly higher within the CD69⁺SLAMF6⁺ precursor exhausted subset. This is intriguing given previous studies have indicated that A_{2A}R signaling is required for the maintenance of naïve cells, partly due to its ability to upregulate IL-7R expression [16, 45-47]. Whether A_{2A}R signaling is required for the maintenance and/or expansion of CD69⁺SLAMF6⁺ precursor exhausted CD8⁺ T cells within tumors remains to be determined but is one potential mechanism by which A_{2A}R signaling may actually promote a favourable differentiation status for responses to immune checkpoint blockade in some contexts.”

4. On our hands, FoxP3⁺ CD4 T cells in the tumors are mostly CD39⁺ or vice versa. And they are co-stained with CD25. A separate staining to confirm this in a naïve mice may convince readers and other reviewers that despite the lack of FoxP3 staining your findings are in fact shows A_{2A}R is very highly expressed in T regs.

Response: To address this we have performed CD25 counterstaining on CD4⁺ T cells isolated from AT3ova tumor bearing mice. This data indicates that the CD4⁺CD39⁺GFP⁺ population contains a significant proportion of CD25⁺ cells whereas CD25 expression is not observed in CD39⁻ counterparts. This new data is presented in New Figure 3G and referred to in the following text on **line 248**.

“This may reflect an increased expression of A_{2A}R on Treg cells, which are known to express high levels of CD39 and PD-1 in the tumor microenvironment [34-37], particularly given CD4⁺GFP⁺CD39⁺ cells were enriched for CD25⁺ cells relative to CD4⁺GFP⁺CD39⁻ counterparts (**Figure 3G**).”

5. Authors states that since A_{2A}R expression in CD8 T cells peaks early and starts getting reduced, treating patients with A_{2A}R antagonists early will cause more benefit. This needs to be supported by data or further speculations/discussions needed such as early A_{2A}R expression may be important for their maintenance, reduction of A_{2A}R later during the growth may mean TME getting less hypoxic (VGEF, vascularization) etc...

Response: To address this we have provided additional data indicating that the subsets of CD8⁺ T cells that express high levels of A_{2A}R diminish in frequency over time. Please refer to point 10 in response to reviewer 1.

6. Making tumors express the cytokine may not be same as treating them. There are reports suggesting MyD88 signaling downstream of IL-1 or IL-18 can be pro-tumoral or anti-tumor depending on the context. IL-18/A_{2A}R hypothesis may better be tested by treating animals with specific antagonists or agonists.

Response: To address this question we have performed new experiments whereby established MC38 tumors were treated with recombinant IL-18 and the expression of A_{2A}R on tumor-infiltrating immune cells determined the next day. Consistent with our results with IL-18 expressing tumor cells we observed a significant increase in A_{2A}R

expression on both CD8⁺ T cells and NK cells. This new data is shown in **New Figure 6J** and referred to in the following text on **line 358** of the revised manuscript.

“Lastly to evaluate the interplay between IL-18 and A2AR expression in a more clinically relevant system we determined the impact of recombinant IL-18 treatment on the expression of A2AR within the context of established MC38 tumors. These experiments revealed that A2AR expression was significantly increased on CD8⁺ T cells and NK cells following treatment with IL-18 (**Figure 6J**).”

7. Alternatively, showing whether combinatorial effect of A2AR/IL-18 is mainly driven by innate or adaptive immunity (by using scid or rag^{-/-} animals) may add value to this study. (Especially given that most of NK cells and activated cDC2s express A2AR)

Response: To address this question we performed either NK cell or CD8 depletion in mice inoculated with either MC38 control or MC38 IL-18 expressing tumors. These results revealed that both NK cells and CD8⁺ T cells contributed to the anti-tumor efficacy of IL-18. This new data is presented in revised **Figure 6I** and referred to in the following text located on **line 357**.

“Depletion experiments revealed that the anti-tumor efficacy of IL-18 was dependent on both CD8⁺ T cells and NK cells (**Figure 6I**).”

8. Discussing the results of this study not only for the pro-tumoral properties of A2AR in general but also anti-tumoral properties especially through increasing metabolic adaptation of T cells in the TME can be fairer. Most antagonists in the trials failed or showed very limited success in mono or combo setting. I think the dynamics of T cell expression of A2AR in TME may be telling something.

Response: We thank the reviewer for this suggestion. This has been addressed with additional discussion as per response to point 3 from reviewer 3 above.

9. The tumor models used in the study are very immunogenic. Testing the dynamics of A2AR expression in tumor models less responsive to IO agents or A2AR blockade (such as B16 melanoma) can add significant value to this study since extremely high or lower than expected expression of A2AR may suggest they are not good targets for these tumors to begin with.

Response: To address this we have performed new experiments in the E0771 model. We have previously shown that this model is non-responsive to immune checkpoint blockade. This new data is presented in **New Supplementary Figure 2**. Expression of A₂AR in this model was comparable to AT3ova and MC38 that we had previously characterized in our original manuscript.

10. Please indicate the unit of tumor size in graphs.

Response: We thank the reviewer for noting this oversight and this has now been corrected.

REVIEWERS' COMMENTS

Reviewer #1 (Remarks to the Author):

The Authors have satisfied all queries by this reviewer and acceptance for publication is supported.

Reviewer #2 (Remarks to the Author):

All my comments have been addressed. I have no additional comments.

Reviewer #3 (Remarks to the Author):

The authors sufficiently addressed my questions and comments from the first round of revision. Their new findings, observations and discussion points are well-aligned with my suggestions.

Point-by-point response to reviewers' comments

Reviewer #1 (Remarks to the Author):

The Authors have satisfied all queries by this reviewer and acceptance for publication is supported.

Reviewer #2 (Remarks to the Author):

All my comments have been addressed. I have no additional comments.

Reviewer #3 (Remarks to the Author):

The authors sufficiently addressed my questions and comments from the first round of revision. Their new findings, observations and discussion points are well-aligned with my suggestions.

Response: We thank each of the reviewers for their positive appraisal of our manuscript and for contributing to the improvement of the final version.